

**Current, steady-state and historical weathering rates of base**
**cations at two forest sites in northern and southern Sweden:**
**A comparison of three methods**
Sophie Casetou-Gustafson[1], Harald Grip[2], Stephen Hillier[3, 5], Sune Linder[4], Bengt A.
Olsson[1], Magnus Simonsson[5] Johan Stendahl[5]
[1]Department of Ecology, Swedish University of Agricultural Sciences, (SLU), P.O. Box 7044, SE-750 07
Uppsala, Sweden
[2]Department of Forest Ecology and Management, SLU, Umeå, Sweden. Present address: Stjernströms väg 5,
SE-129 35 Hägersten, Sweden
[3]The James Hutton Institute, Craigiebuckler, Aberdeen AB15 8QH, United Kingdom
[4]Southern Swedish Forest Research Centre, SLU, P.O. Box 49, SE-230 53 Alnarp, Sweden
[5]Department of Soil and Environment, SLU, P.O. Box 7014, SE-750 07 Uppsala, Sweden
*Correspondence to*: Sophie Casetou-Gustafson (Sophie.Casetou@slu.se)

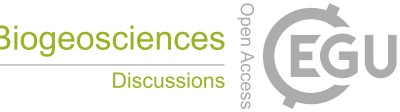


**Abstract**
Reliable and accurate methods for estimating soil mineral weathering rate are required tools in evaluating the
sustainability of increased harvesting of forest biomass. A variety of methods that differ in concept, temporal and
spatial scale and data requirements are available for measuring weathering rate. In this study, release rates of base
cations through weathering were estimated in podsolised glacial tills at two experimental forest sites, Asa and
Flakaliden, in southern and northern Sweden, respectively. Three different methods were used: (i) historical
weathering since deglaciation estimated with the depletion method, using Zr as assumed inert reference; (ii)
steady-state weathering rate estimated with the PROFILE model, based on quantitative analysis of soil
mineralogy; and (iii) base cation mass balance at stand scale, using measured deposition, leaching and changes in
base cation stocks in biomass and soil over a period of 12 years.

36           In the 0-50 cm soil layer at Asa, historical weathering of Ca, Mg, K and Na estimated by the depletion

method was 4.7, 3.1, 0.8 and 2.0 mmol$_c$ m$^{-2}$ yr$^{-1}$, respectively. Corresponding values at Flakaliden were 7.3, 9.0,
1.7 and 4.4 mmol$_c$ m$^{-2}$ yr$^{-1}$, respectively. Steady state weathering rate for Ca, Mg, K and Na estimated with
PROFILE was 8.9, 3.8, 5.9 and 18.5 mmol$_c$ m$^{-2}$ yr$^{-1}$, respectively, at Asa and 11.9, 6.7, 6.6 and 17.5 mmol$_c$ m$^{-2}$
yr$^{-1}$, respectively, at Flakaliden. Thus at both sites, the PROFILE results indicated that steady-state weathering
rate increased with soil depth as a function of exposed mineral surface area, reaching a maximum rate at 80 cm
(Asa) and 60 cm (Flakaliden). In contrast, the depletion method indicated that the largest postglacial losses were
in upper soil layers, particularly at Flakaliden.

44           With the exception of Mg and Ca in shallow soil layers, PROFILE appeared to produce consistently higher

weathering rates, particularly of K and Na in deeper soil layers. In contrast, the depletion method appeared to to
produce consistently lower rather than higher weathering rates, due to natural and anthropogenic variability in
(reference) Zr gradients. The mass balance approach produced significantly higher weathering rates of Ca, Mg,
and K (65, 23, 40 mmol$_c$ m$^{-2}$ yr$^{-1}$ at Asa and 35, 14 and 22 mmol$_c$ m$^{-2}$ yr$^{-1}$ at Flakaliden), but lower Na weathering
rates similar to the depletion method (6.6 and 2.2 mmol$_c$ m$^{-2}$ yr$^{-1}$ at Asa and Flakaliden). The large discrepancy in
weathering rates for Ca, Mg and K between mass balance and the other methods suggest that there were additional
sources for tree uptake in the soil besides weathering and measured depletion in exchangeable base cations.

**Keywords**. Weathering; minerals; soil layers; nutrient mass-balance; *Picea abies*; PROFILE model; depletion;
mass balance approach









**Definitions and abbreviations**


Mineralogy = The identity and stoichiometry of minerals present in a certain geographical unit, a particular site
(*site-specific mineralogy*) or a larger geographical province (*regional mineralogy*)
Quantitative mineralogy or mineral composition = Quantitative information (wt.%) on the abundance of specific
minerals in the soil.
Weathering rate = Weathering of a mineral resulting in release of a base cation.
**Abbreviations:**
$W_{depletion}$ = Historical weathering rate based on calculation of loss of mobile elements since last deglaciation
$W_{profile}$ = Steady-state weathering rate estimated using the PROFILE model
$W_{mb}$ = Current weathering rate based on mass balance calculations


















## 1. Introduction

Silicate weathering is the major long-term source of base cations in forest ecosystems (Sverdrup et al., 1988) and is therefore crucial for sustainable plant production and for proton consumption, counteracting soil and water acidification (Nilsson et al., 1982; Hedin et al., 1994; Likens et al., 1998; Bailey et al., 2003). These effects of weathering are important in areas where historical high sulphur deposition (S) causes severe acidification of forest soils and waters (Reuss and Johnson, 1986), particularly in southern Scandinavia where acid silicate bedrock and less readily weatherable soils are abundant (Likens and Bormann, 1974). By 1990 in most European countires, the trend of increasing S emissions since the 1950s started to abate (Grennfelt and Hov, 2005) and recovery from acidification began (Warfvinge and Bertills; 2000; Bertills et al., 2007). This recovery is mainly driven by silicate weathering (Evans et al., 2001; Skjelkvåle et al., 2001), but the process is slow (Akselsson et al., 2013; Futter et al., 2012). At the same time, forest growth, has become a more important source of acidity caused by the accumulation of base cations in tree biomass in excess over anion uptake (Nilsson et al., 1982). The biomass extraction rate at harvesting determines the extent to which soil acidity produced by forest growth can be neutralised by decomposition of biomass left on-site. Whole-tree harvesting can thus result in more acid, base cation-depleted soils than stem-only harvesting (Olsson et al., 1996; Zetterberg et al., 2013), due to smaller return of alkalinity. The combined effect of increased productivity of forests in Sweden, resulting in increased stocks of forest biomass, and increased use of whole-tree harvesting for energy purposes can therefore impede recovery from acidification and place increasing demands on nutrient supply. The current Swedish environmental quality objective "Natural Acidification Only" targets a reduction in acid load due to forest growth and harvesting and to acid deposition (Bertills et al., 2007).

The role of weathering in maintaining base cation balance in Swedish forest soils has been examined in several previous studies (Sverdrup and Rosén, 1998; Akselsson et al., 2007). A regional-scale study on Swedish forest soils found that, in parts of Sweden, base cation depletion can occur at rates that lead to negative effects on e.g. soil fertility and runoff water quality within one forest rotation (Akselsson et al., 2007). The methods used to determine base cation balance in that study included predicting weathering rate based on the PROFILE model and analysing regional data for deposition, leaching and base cation losses in harvested biomass. Negative balances, which indicate soil base cation depletion, were found to be more frequent after whole-tree harvesting than stem-only harvesting, especially for Norway spruce compared with Scots pine, with the effect being more common in southern than in northern (boreal) Sweden. Among the elements studied, Ca was most frequently subject to depletion. However, uncertainties in estimating the terms of the base cation balance can accumulate to produce large uncertainties in the overall balance, and therefore it is difficult to draw firm conclusions about the sustainability of different harvesting options based on base cation balance alone. There are also conflicting opinions about the consequences of long-term negative base cation balance in soils for sustainable forest production (Binkley and Högberg, 2016). Experimental studies have demonstrated that growth of boreal forests is strongly limited by nitrogen (N), whereas co-limitation with phosphorus (P) may occur in hemiboreal and temperate forests (e.g. Tamm, 1991). Thus, concerns about base cation depletion in Swedish forest soils following increasing use of whole-tree harvesting have so far been about soil acidification rather than tree nutrition,





expressed for example by the Swedish Forest Agency (2008) in their recommendation on nutrient compensation
with wood ash.
In regional assessments of the sustainability of different harvesting regimes, the value of weathering rate used has
a strong influence. Klaminder et al. (2011b) found that different approaches to estimating weathering rates yielded
results that differed substantially, and that uncertainties in the methods had a great influence on the predicted
sustainability of different harvesting practices.
Futter et al. (2012) compiled weathering rates estimated at 82 sites, using different methods, and found both large
between-site as well as within-site differences in the values. They concluded that the sources of uncertainties were
as follows: input data > parameters > weathering concepts/assumptions. Differences in input data can be attributed
to different time scales, challenges determining accurate mineralogical compositions and the use of laboratory
data compared with field data (Van der Salm, 2001; Klaminder et al., 2011b). Thus, they recommend that at least
three different approaches be applied per study site to increase the precision in weathering estimates.
Beside the PROFILE model, the depletion method is used in Sweden to estimate weathering rates at regional scale
(Olsson et al., 1993). The depletion method estimates total base cation losses since deglaciation in the soil above
a reference soil depth, using an element in a weathering-resistant mineral as a standard, commonly zirconium (Zr,
present in zircon) or titanium (Ti, present in rutile etc.) (Sudom and St.Arnaud, 1971; Harden et al., 1987;
Chadwick et al., 1990; Bain et al., 1994), due to their stability at low temperatures (Schützel et al., 1963). To yield
an annual average weathering ($mmol_c$ $m^{-2}$), calculated element losses are commonly divided by an estimated
number of years lapsed since the onset of weathering. Steady-state weathering rate may differ from the average
during soil formation, which is one reason why weathering rates obtained using PROFILE and the variation in
rate with depth in the soil can be expected to differ from those obtained using the depletion method (Stendahl et
al., 2013). Observed discrepancies between these methods may therefore reflect 'true' differences or conceptual
differences between the methods.

The mechanistic PROFILE model estimates steady state release rates of base cations based on the dissolution
kinetics of a user-defined set of minerals in the soil and the physical and chemical conditions that drive dissolution
of minerals. Since it is a mechanistic model, its strength is its transparency, while its main weakness is the
difficulty in setting values of model parameters and input variables to which it may have high sensitivity.
Akselsson et al. (*this issue*) concluded that the most important way to reduce uncertainties in modelled weathering
rates is to reduce input data uncertainties, e.g. regarding soil texture, although there is also still room for
improvement in process descriptions of e.g. biological weathering and weathering brakes (e.g. Lampa Erlandsson
et al., *this issue*). The sensitivity of PROFILE to variations in soil physical parameters (e.g. soil texture, soil bulk
density) and mineral composition is discussed by Jönsson et al. (1995) and Hodson et al. (1996), while the
importance of the ability to determine the identity and quantity of the minerals is analysed by Casetou-Gustafson
et al. (*this issue*).



An alternative approach to estimating weathering rate is the mass balance method (Likens et al., 1998; Velbel,
1985). It has been applied to estimate current weathering rates at various temporal and spatial scales, and elements
of mass balance have been used in different ways in some popular models, e.g. MAGIC (Cosby et al., 1985).
Using the mass balance approach, weathering rate is estimated from the difference between sinks and sources of
elements within a system with defined boundaries. The missing source in the mass balance equation is assumed
to contain the weathering, but can also contain other unidentified sources.  The mass balance approach is most
reliable when based on long-term data from well-defined systems, although estimates of weathering tend to suffer
from large uncertainties, as errors can be expected to accumulate in the mass balance equation (Simonsson et al.,
2015). Furthermore, the mass balance approach has rarely been applied under non-steady state conditions (i.e.
including measures of soil exchangeable pools), due to lack of long-term data on base cation fluxes. Consequently,
at the pedon scale, the PROFILE model and the depletion method are the most frequently used methods in Sweden
to estimate weathering rate. The benefit of comparing these two methods is that, taken together, they can provide
complementary information about soil weathering potential (i.e. historical versus steady state weathering) in
individual soil layers.
The main aim of this study was to quantify weathering rates in two young Norway spruce forests using the
depletion method and the PROFILE model and to examine possible causes of discrepancies. A second aim was
to quantify weathering rates by the mass balance method and to compare weathering estimates obtained by the
different methods against other major base cation fluxes at ecosystem scale.
The different methods were applied at two ecosystem experiment sites in southern (Asa) and northern (Flakaliden)
Sweden, where available data allowed construction of base cation mass balances at stand scale. These study sites
are representative of Norway spruce forest in Sweden on soil derived from glacial till soil derived from mostly
acid silicate rock. They have similar soil texture and structure, but differ in climate and pedogenesis.
Three test criteria were used to examine the outputs of the methods: (1) similarity in weathering estimates for the
0-50 cm soil profile; (2) similarity in depth gradients in weathering; and (3) similarity in ranking order of the base
cations released.
**2. Materials and methods**
**2. 1 Study sites**
Two forest sites planted with Norway spruce (*Picea abies* (L.) Karst) were chosen for the study, Flakaliden in
northern Sweden (64°07'N, 19°27'E) and Asa in southern Sweden (57°08'N, 14°45'E), because they have been
used for long-term experimental studies on the effects of climate and nutrient and water supply on tree structure
and function and element cycling (Linder, 1995; Bergh et al., 1999; Ryan, 2013).
The experiment at Flakaliden was established in 1986 in a 23-year-old Norway spruce stand, planted in 1963 with
four-year-old seedlings of local provenance after prescribed burning and soil scarification (Bergh et al., 1999).



The experiment at Asa was established one year later (1987), in a 12-year-old Norway spruce stand planted in
1975 with two-year-old seedlings after clear-felling and soil scarification. The experimental design was similar at
both sites and included control, irrigation and two nutrient optimisation treatments (Bergh et al., 1999). All
treatments had four replicate 50 m x 50 m plots, arranged in a randomised block design. Only two of the four
treatments were used in the present study; the control (C) and plots receiving an annual dose of an optimised mix
of solid fertiliser (F). For further details, see Linder (1995).
Flakaliden is located in the central boreal sub-zone with a harsh climate, with long cool days in summer and short
cold days in winter. Mean annual temperature for the period 1990-2009 was 2.5 °C, and mean monthly
temperature varied from -7.5 °C in February to 14.5 °C in July. Mean annual precipitation in the period was ~650
mm, with approximately one-third falling as snow, which usually covers the frozen ground from mid-October to
early May. Mean length of the growing season (daily mean temperature ≥ 5 °C) was 148 days, but with large
between-year variations (Table 1) (cf. Sigurdsson et al., 2013).
Asa is located in the hemi-boreal zone, where the climate is milder than at Flakaliden, which is reflected in a
longer growing season (193 days). Mean annual temperature (1990-2009) was 6.3 °C, mean monthly temperature
varied from -1.9 °C in February to 16.0 °C in July and mean annual precipitation was ~750 mm. The soil is
periodically frozen in winter. The difference in climate is reflected in differences in site productivity, which
broadly follows climate gradients in Sweden (Bergh et al., 2005).
The soils at Asa and Flakaliden differ in age due to differences in the time since deglaciation (Table 1). Soil age
is approximately 143 000 thousand years at Asa and 10 150 thousand years at Flakaliden (based on National Atlas
of Sweden; Fredén, 2009). The soil type at both sites is Udic Spodosol, with a mor humus layer overlying acid
silicate bedrock. The soil texture is classified as sandy loamy till. The B-horizon transitions to C-horizon at 35-
40 cm depth at Flakaliden and 40 cm depth at Asa. The natural ground vegetation at Flakaliden is dominated by
*Vaccinium myrtillus* (L.) and *V. vitis-idaea* (L.) dwarf-shrubs, lichens and mosses (Kellner, 1993; Strengbom et
al., 2011), while the ground vegetation at Asa is dominated by *Deschampisa flexuosa,* (L.) and mosses (Strengbom
et al., 2011; Hedwall et al., 2013).
**2.2. Soil sampling and analyses of geochemistry and mineralogy**
A detailed description of soil sampling, geochemical analyses and determination of mineralogy can be found in
Casetou-Gustafson et al. (2018). The procedures are summarised below.
Soil sampling was performed in October 2013 (Flakaliden) and March 2014 (Asa), in the border zone of four plots
at each site. Plots selected for sampling were untreated control plots (K1 and K4 at Asa; 10B and 14B at
Flakaliden) and fertilised (F) plots (F3, F4 at Asa; 15A, 11B at Flakaliden). A rotary drill (17 cm inner diameter)
was used to extract one intact soil core per plot at Flakaliden and in plot K1 at Asa. In plots K4, F3 and F4 at Asa,
soil samples were taken from the wall of 1 m deep soil pits, due to inaccessible terrain for forest machinery.
Maximum soil depth was shallower at Flakaliden (70-90 cm) than at Asa (90-100 cm). The volume of stones and



boulders was determined for each plot at the two study sites using the penetration method described by Viro
(1952) to a maximum depth of 30 cm and by applying the fitted function described by Stendahl et al. (2009).
Mean stone and bolder content was higher at Flakaliden (39%$_{vol}$) than at Asa (28%$_{vol}$).
Soil samples were taken from each 10-cm soil layer. Prior to chemical analysis, these samples were dried at 30-
40 °C and sieved (2 mm mesh). Particle size distribution was analysed by wet sieving and sedimentation (pipette
method) in accordance with ISO 11277. Geochemical analyses were conducted by ALS Scandinavia AB and
comprised inductively coupled plasma-mass spectrometry (ICP-MS) on HNO$_3$ extracts of fused samples that were
milled and ignited (1000 °C) prior to fusion with LiBO$_2$.
Quantitative soil mineralogy was determined with the X-ray powder diffraction technique (XRPD) (Hiller 1999,
2003). Samples for determination of XRPD patterns were prepared by spraying and drying slurries of micronised
soil samples (<2 mm) in ethanol. Quantitative mineralogical analysis of the diffraction data was performed using
a full pattern fitting approach (Omotoso et al., 2006). In this fitting process, the measured diffraction pattern is
modelled as a weighted sum of previously verified standard reference patterns of the previously identified mineral
components. The chemical composition of the various minerals present in the soils was determined by electron
microprobe analysis (EMPA) of mineral grains subsampled from the sifted (< 2 mm) soil samples.
**2.3 Historical weathering determined with the depletion method**
**2.3.1 Method description**
The depletion method, as defined by Marshall and Haseman (1943) and Brimhall et al. (1991), estimates the
accumulated mass loss since soil formation (last deglaciation for our sites) as a function of loss of a mobile
(weatherable) element and enrichment of an immobile (weathering resistant) element. Zirconium is commonly
used as the immobile element due to the inert nature of the mineral zircon (ZrSiO$_4$), although Ti is sometimes
used due to the resistance of the Ti-containing minerals anastase/rutile (TiO$_2$) to weathering (Olsson and
Melkerud, 1989). The maximum weathering depth, below which weathering is assumed to cease, is defined by a
reference layer andabove the reference layer, Zr is enriched compared with other elements (i.e. base cations). The
method is based on the assumptions that zircon was uniformly distributed throughout the soil profile at the time
of deglaciation, that weathering only occurs above the reference layer and that zircon does not weather. The latter
implies that Zr gradient and Zr/BC ratio are constant below the reference layer. Table 2 shows the reference depths
for different base cations compared with Zr, which were used as the depths of immobile element concentrations.
**2.3.2. Assumption testing**
Prior to calculating base cation weathering rates with the depletion method, the homogeneity of the parent material
was evaluated (Fig. 1). Since zircon and anastase/rutile are weathering resistant minerals, it was assumed that the
ratio of Ti to Zr would be more or less stable as soil depth increases and, as such, uniformity of the parent material
could be ensured. Use of the ratio of two resistant minerals to establish uniformity of parent material has been
suggested previously (Sudom and St.Arnaud, 1971; Starr et al., 2014). This was the case for plots F3, K4 and F4
at Asa, but there was somewhat more variability in plot K1 (i.e. the Zr concentrations decreased towards the soil





surface; Fig. 5), consequently Ti was used as the immobile element instead of Zr for this profile. At Flakaliden,
variability in both the Zr and Ti gradients was observed, but the parent material was considered sufficiently
uniform for all plots except 15A.

**2.3.3. Input data**

Bulk density was estimated for each soil layer except in some plots where density measurements could not be
made below a certain soil depth or where a large and sudden decrease in bulk density with increasing soil depth
was observed. Bulk density in these cases was estimated using an exponential model for total organic carbon
(TOC) and bulk density (BD, g/cm$^3$) based on our own data. For Asa (F3: soil layer 70-90 cm; F4: 0-10, 30-40,
50-60, 60-70, 70-80, 80-90, 90-100 cm; K4: 70-80, 80-90, 90-100 cm), the following function was used:
$BD = 1.3\ e^{-0.1x}$ (1)
where x is TOC content (% of dry matter).
For Flakaliden (14B: 80-90cm; 10B: 60-70 cm; 11B:40-70 cm), the function used was:
$BD = 1.8\ e^{-0.2x}$ (2)

**2.4 The PROFILE model**

**2.4.1 Model description**

The biogeochemical PROFILE model can be used to estimate the steady state weathering of soil profiles, as
weathering is known to be primarily determined by soil physical properties at the interface of wetted mineral
surfaces and the soil solution. PROFILE is a multilayer model, where parameters are specified for each soil layer
based on field measurements and estimation methods (Warfvinge and Sverdrup, 1995).

**2.4.2 PROFILE parameter estimation**

A detailed description of application of the PROFILE model to the soils and sites in the present study can be found
in Casetou-Gustafson et al. (*this issue*). The parameters used are listed in Table 3.
Exposed mineral surface areas were estimated from soil bulk density and texture data using the algorithm specified
in Warfvinge and Sverdrup (1995). Volumetric field soil water content for each soil pit in Flakaliden and Asa was
estimated to be 0.25 m$^3$ m$^{-3}$ according to the moisture classification scheme described in Warfvinge and Sverdrup

292 (1995).

Aluminium (Al) solubility coefficient, a soil chemical parameter needed for solution equilibrium reactions, was
defined as $\log\{Al^{3+}\}+3pH$. It was estimated by applying a function developed from previously published data
(Simonsson and Berggren, 1998) and existing total carbon and oxalate-extractable Al measurements for our sites
(Casetou-Gustafson et al., 2018). For partial $CO_2$ pressure in the soil, the default value of Warfvinge and Sverdrup
(1995) was used. Data on measured dissolved organic carbon (DOC) in the soil solution at 50 cm depth were
available for plots K4 and K1 at Asa and plots 10B and 14B at Flakaliden, and these values were also applied for
deeper soil horizons. Shallower horizons (0-50 cm) were characterised by higher DOC values, based on previous
findings (Fröberg et al., 2013) and the DOC classification scheme in Warfvinge and Sverdrup (1995).



The site-specific parameters used were evapotranspiration, temperature, atmospheric deposition, precipitation, runoff and nutrient uptake in biomass. Mean evapotranspiration per site was estimated from mean annual precipitation and runoff data, using a general water balance equation.

Total deposition was calculated using deposition data from two sites of the Swedish ICP Integrated Monitoring catchments, Aneboda (for Asa) and Gammtratten (for Flakaliden) (Löfgren et al., 2011). The canopy budget method of Staelens et al. (2008) was applied as in Zetterberg et al. (2016). The canopy budget model is commonly used for elements that are prone to canopy leaching ($Ca^{2+}$, $Mg^{2+}$, $K^+$, $Na^+$, $SO_4^{2-}$) or canopy uptake ($NH_4^+$, $NO_3^-$) and it calculates total deposition as the sum of dry deposition and wet deposition. Wet deposition was estimated based on the contribution of dry deposition to bulk deposition, both for base cations and anions, using dry deposition factors from Karlsson et al. (2012,2013).

Net base cation and nitrogen uptake in aboveground tree biomass (i.e. bark, stemwood, living and dead branches, needles) was estimated as mean accumulation rate over a 100-year rotation period in Flakaliden and a 73-year rotation period in Asa. These calculations were based on Heureka simulations using the StandWise application (Wikström et al., 2011) for biomass estimates, in combination with measured nutrient concentrations in aboveground biomass (Linder, unpubl. data).

### 2.4.3 PROFILE sensitivity analysis

The sensitivity of PROFILE to a change in soil physical and mineralogical input was analysed, to test whether depth gradients of weathering rates predicted by PROFILE could be explained by either soil physical properties or soil mineralogy. Independent PROFILE runs were performed for each of the following three test scenarios: (1) homogeneous soil physical properties (soil bulk density and specific exposed mineral surface area); (2) homogeneous mineralogy; and (3) homogeneous soil (i.e. soil physics and mineralogy). For homogeneous soil, all soil layers (0-100 cm) were given the same value of the test variable, i.e. the average value for the actual soil profile (0-100 cm).

### 2.5 The mass balance method

### 2.5.1. General concepts of the mass balance method

The average weathering rate of base cations ($W_{BC}$) over a period of time can be estimated using the mass balance approach, which assumes that total deposition ($TD_{BC}$) and weathering are the major sources of mobile and plant-available base cations in the soil, and that leaching ($L_{BC}$) and accumulation of base cations in biomass ($B_{BC}$) are the major sinks. A change in the extractable soil stocks of base cations over time ($\Delta S_{BC}$) can be considered as a sink if stocks have increased, or as a source if stocks have been depleted (Simonsson et al., 2015). Each of these terms is measured independently over a specific period of time. Hence,

$$W_{BC} = L_{BC} + B_{BC} + \Delta S_{BC} - TD_{BC} \tag{3}$$

### 2.5.2 Atmospheric deposition, $TD_{BC}$



The same estimates of total atmospheric deposition as used in parameter setting of the PROFILE model (section
2.4.2) were used in the mass balance equation 3.

### 2.5.3. Changes in exchangeable soil pools, $\Delta S_{BC}$

Changes in extractable base cation stocks in the soil were calculated from linear regressions of stocks measured
at repeated soil samplings. The organic layer and the mineral soil were sampled to 40 cm or deeper in 1986 and
1998 at Flakaliden, and in 1988 and 2004 at Asa.
At Flakaliden, soil sampling for chemical analyses was carried out in September 1986 in the border zone of plots,
using a 5.6 cm diameter corer for the organic layer and a 2.5 cm diameter corer for the mineral soil. The plots
were re-sampled in 1998 for chemical analyses, in the same way as in 1986. Soil sampling at Asa was conducted
in 1988, prior to the start of the experimental treatments, using the same method as at Flakaliden. The transition
from the humus layer to the mineral soil was less clear at Asa than at Flakaliden.
Exchangeable base cation content in the soil (<2 mm) in the Flakaliden samples from 1986 and 1998 and the Asa
samples from 1988 was determined by extraction of dry samples with 1 M $NH_4Cl$ using a percolation method,
where 100 mL of solution was percolated through 2.5 g of sample at a rate of around 20 mL $h^{-1}$. The base cations
were analysed by atomic absorption spectrophotometry (AAS). For the Asa samples from 2004, extraction was
performed using the same extractant in a batch extraction method and the base cations were determined with ICP.
A separate test was made to compare the yield of the percolation and batch extraction methods, but the results
were inconclusive and therefore no correction was made to account for possible differences between the extraction
methods.
The amount of fine soil (<2 mm) per unit area was calculated from the volume of fine soil in the soil profiles and
the average bulk density of the soil in the 0-10, 10-20 and 20-40 cm layers. Bulk density and volume proportion
of stoniness at Flakaliden were determined from sampling in 1986 in 20 soil profiles (0.5 m x 0.5 m and about 0.5
m deep) outside plots. At Asa, stoniness was determined with the penetration method of Stendahl et al. (2009)
and the bulk density of soil <2 mm was calculated using a pedotransfer function that included soil depth and
measured carbon concentrations as variables.

### 2.5.4 Net uptake in biomass, $B_{BC}$ (1987-2003)

Accumulation of base cations in tree biomass, i.e. net uptake of base cations, was calculated as mean value of
control plots over the period 1989-2003. The calculations were based on increments in aboveground biomass at
Asa and Flakaliden for this period (Albaugh et al., 2009, 2012) and concentrations of elements in different tree
parts. The increment in belowground biomass was estimated from general allometric functions for Norway spruce
stumps and roots in Sweden (Marklund, 1988). Since Marklund's functions (1988) underestimate belowground
biomass by 11%, a factor to correct for this was included (Petersson and Ståhl, 2006). Furthermore, the finest root
fraction (≤2 mm), which is not included in the functions of Marklund (1988) and Petersson and Ståhl (2006), was



assumed to be 20% of needle biomass at Asa and 40% at Flakaliden, respectively, based on data from Helmisaari
et al. (2007).

Data on element concentrations in biomass were available from measurements on harvested trees (S. Linder,
unpubl. data). At Flakaliden, total element concentrations were analysed in trees sampled for biomass
determination in 1992 and 1997. Needles and branches (dead and live) were conducted on the same tree parts in
the biomass sampled in 1993 and 1998. Base cation concentrations in biomass were determined from acid wet
digestion in $HNO_3$ and $HClO_4$, followed by determination of elements by ICP-atomic emission spectrophotometry
(ICP-AES) (Jobin Yvon JY-70 Plus).

Data on element concentrations in belowground biomass fractions were taken from literature from the Nordic
countries (Hellsten et al., 2013). Specifically, data on stump and root biomass of Norway spruce were available
for Asa and data from Svartberget was used for Flakaliden (Table 7 in Hellsten et al., 2013).
**2.5.5. Leaching, $L_{BC}$**
Base cation leaching was quantified in 6-month intervals from modelled runoff multiplied by average element
concentrations in soil water collected with tension lysimeters at 50 cm soil depth. Runoff was calculated using
CoupModel (Jansson, 2012).

Annual precipitation varied considerably during the period 1990-2002, ranging from 906 to 504 mm at Flakaliden
(mean 649 mm) and from 888 to 575 mm at Asa (mean 736 mm).

CoupModel was parameterised based on measured hydraulic soil properties. The model was run with measured
climate variables (global radiation, wind speed, air temperature and humidity) and model outcomes were tested
against tensiometer data. The parameters were then adjusted slightly to obtain the best agreement between
measured and calculated soil water content. Annual evapotranspiration increased by about 50 mm at both sites,
during the period 1987-2003 at Flakaliden and 1990-2002 at Asa, due to the increment in tree biomass. Soil water
was collected from five ceramic tension lysimeters (P80) installed at 50 cm depth in each experimental plot. Soil
water was collected on 2-3 occasions per year during frost-free seasons, applying an initial tension of 70 kPa in
250 mL sampling bottles, and left overnight. These soil water samples were pooled by plot. The base cation
concentration in the soil solution was determined with ICP-AES.
**2.5.6. Judgement of data quality in mass balance**
The precision and accuracy of a mass balance estimate of $W_{BC}$ is determined by the quality of estimates of each
individual term in equation 3, in proportion to the magnitude of each term (Simonsson et al., 2015). Significant
uncertainty in the estimate of a quantitatively important term will therefore dominate the overall uncertainty in
estimates of $W_{BC}$. The quality of data for each term in equation 3 was assessed based on the spatial and temporal
scales of measurements and the quality of measurements (Table 4). Based on these criteria, we consider the
estimates of deposition, leaching and accumulation in biomass to be of moderate to high quality. The





measurements of changes in extractable soil pools were of lower quality because of bias in methods in repeated
samplings, which would cause significant uncertainty if soil changes were an important part of the element budget.
To partly overcome this uncertainty, we used the estimates of $W_{BC}$ obtained by the PROFILE and depletion
methods in additional mass balance calculations where the change in soil was determined from the mass balance.
These additional mass balance estimates, which are conceptually analogous to the regional mass balances
presented by Akselsson et al. (2007), were also used to place the PROFILE and depletion method estimates of
$W_{BC}$ in the context of other base cation fluxes at ecosystem scale.
**2.6 Statistical analyses**
Site mean values and standard error (SE) of weathering rates were calculated based on the four soil profiles studied
at each site ($W_{depletion}$, $W_{profile}$) and on the $W_{MB}$ estimates for the four control plots at each site. R (version 3.3.0)
(R Core Team, 2016) and Excel 2016 were used for statistical plotting of results..

**3. Results**
**3.1 Depletion method estimates of historical weathering rates**
At both Asa and Flakaliden, historical weathering rates estimated with the depletion method ($W_{depletion}$) were
highest in the upper soil layers and showed a gradual decrease down to the reference depth, which was defined in
most plots at 60-70 cm at Flakaliden and at 80-90 cm at Asa (Fig. 2). Weathering had also taken place below the
reference depth. In line with the younger age of the soils at Flakaliden (indicated also by higher abundance of the
more easily weatehrbale minerals amphibole, trioctahedral phyllosilicates and calcic plagioclase), the historical
annual weathering rate was higher at Flakaliden compared to Asa. The base cation weathering down to 90 cm soil
depth amounted to 12.8 mmol$_c$ m$^{-2}$yr$^{-1}$ at Asa and 25.1 mmol$_c$ m$^{-2}$yr$^{-1}$ at Flakaliden. The corresponding value for
the 0-50 cm horizon was 10.5 mmol$_c$ m$^{-2}$yr$^{-1}$ at Asa and 22.4 mmol$_c$ m$^{-2}$yr$^{-1}$ at Flakaliden. The gradients with depth
showed that $W_{depletion}$ increased towards the surface, although this trend was more pronounced at Flakaliden than
at Asa. Furthermore, at Flakaliden, $W_{depletion}$ was highest for Mg, followed by Ca, Na and K (Figs. 2 and 3). At
Asa, the largest average mass loss was observed for Ca, closely followed by Mg, Na and K (Figs. 2 and 3).
**3.2 PROFILE model estimates of steady state weathering rates**
The steady state weathering rate estimated by the PROFILE model ($W_{profile}$) differed from the historical rate with
respect to all aspects covered by the three starting hypotheses, i.e. total weathering rate in the 0-50 cm soil layer,
variation in weathering with depth and ranking order of base cations (Figs. 2 and 3).
PROFILE predicted that weathering rates increased with soil depth at Asa (down to 90 cm) and Flakaliden (down
to 60 cm). At Flakaliden, anomalously high contents of K- and Mg-bearing tri-octahedral mica (Casetou-
Gustafson et al., 2018) gave rise to increased weathering rates at 70-80 cm. Apart from in that specific soil layer,
at both sites $W_{profile}$ was largest for Na, followed by Ca. However, $W_{profile}$ was larger for K than for Mg at Asa,
while the reverse was true at Flakaliden.





In general, PROFILE also predicted much higher weathering rates than the depletion method (Fig. 2). However, both methods estimated consistently higher weathering rates at Flakaliden than Asa. The total modelled base cation weathering rate for the soil profile down to 90 cm was around 7-fold higher than the rate estimated using the depletion method at Asa (89.4 mmol$_c$ m$^{-2}$yr$^{-1}$), and almost 5-fold higher at Flakaliden (127.6 mmol$_c$ m$^{-2}$yr$^{-1}$).

Weathering rates are often reported for the upper 50 cm soil layer, as an approximation of the root zone. On restricting the base cation weathering rate to the upper 50 cm of the mineral soil, W$_{profile}$ estimates for Asa and Flakaliden were more similar (Asa: 37.1 mmol$_c$ m$^{-2}$yr$^{-1}$, Flakaliden: 42.7 mmol$_c$ m$^{-2}$yr$^{-1}$). This was not the case for W$_{depletion}$ as shown above, as the estimate for Flakaliden was more than twice that obtained for Asa. However, W$_{depletion}$ was higher than W$_{profile}$ for Mg. In relative terms, the difference between sites in W$_{depletion}$ in the 0-50 cm layer was similar to the difference observed for the whole soil profile.

The sensitivity analysis of the PROFILE model using homogeneous soil physical and/or mineralogical properties demonstrated that variation in soil physical parameters (i.e. soil texture and density) with depth, rather than mineralogy, was the most important input data explaining the observed change in W$_{profile}$ with soil depth. Applying homogeneous mineralogy had little effect with the original gradient of W$_{profile}$ (Fig. 2) being similarly reproduced with depth.However, when homogenous soil physical conditions were applied (i.e. a combination of homogeneity in soil physics and soil mineralogy), there were some small variations in W$_{profile}$ between soil layers (Tables S1 and S2). In terms of sum of squared error, calculated from the difference in W$_{profile}$ between actual and homogeneous soil, 75-85% of the total error at Flakaliden was due to homogeneity in soil physics only, while the error due to homogeneous mineralogy was 2-17%. The corresponding error values for Asa were 76-94% and 1-4%, respectively. One exception to these findings was plot 10B at Flakaliden, which showed somewhat higher error (17%) resulting from homogeneous mineralogy. This was due to the anomalously high content of trioctahedral micas at 70-80 cm depth, as previously reported by Casetou-Gustafson et al. (2018). At Asa, an outlier to the results was plot F3, where homogeneous soil physical properties produced 109% of the error resulting from homogeneous soil. This was because, by coincidence, soil profile F3 had relatively high bulk density and exposed mineral surface area in the uppermost soil layer, compared with the average soil physical input data, and therefore the homogeneous soil test produced lower weathering rates. Soil physical input parameters that were more important for PROFILE weathering rates are indicated in Figs. S1 and S2. There was a strong linear and positive relationship between exposed mineral surface area and W$_{profile}$ for all elements at both sites, with R$^2$ values ranging from 0.65 to 0.89 (Fig. S1). The relationship between bulk density and W$_{profile}$ was also strong and showed the same linear response, although R$^2$ values were lower, 0.40-0.70 (Fig. S2).

**3.3 Mass balance estimates of current weathering rates**

A comparison of weathering rates estimated by mass balance (W$_{mb}$), W$_{profile}$ and W$_{depletion}$ was made for the 0-50 cm soil layer. The physical boundary for the mass balance was defined by the depth of soil solution sampling (50 cm). It was found that, for most elements, W$_{mb}$ in the 0-50 cm layer was higher, or much higher, than W$_{profile}$ (Fig. 3). Compared with the PROFILE model estimates, the mass balance estimates of weathering were 6- to 7-fold higher for Ca, Mg and K weathering at Asa, and about 2- to 3-fold higher for Ca, Mg and K at Flakaliden. At Asa,



the sum of base cations was on average 13-fold and 3.6-fold larger than average annual long-term weathering
rates based on the depletion method and PROFILE method, respectively. The closest fit between methods was
found between $W_{depletion}$ and $W_{mb}$ for Na.
**3.4 Base cation fluxes in measured mass balances**
A general pattern in base cation fluxes in the mass balances was the difference between Na, on the one hand, and
K, Mg and Ca on the other (Fig. 4). This difference was largely due to accumulation in biomass being the dominant
sink for the latter elements, whereas Na uptake in biomass was negligible and leaching was the dominant sink.
Compared with biomass uptake, loss by leaching was a negligible sink for K, but a significant sink for Mg and
Na.

Deposition was a minor source of base cation inputs, except for Na at Asa. The measured decreases in soil stocks
of exchangeable base cations indicated that a change in this pool was a particularly important source of Ca. There
were minor increases in exchangeable stocks for Na, K and Mg at Asa.

In summary, the mass balance calculations indicated that weathering was a particularly dominant source of K and
Mg, but weathering contributed a relatively smaller proportion of the total Ca sources than for K and Mg (61% at
Asa and 43% at Flakaliden).

By using the weathering estimates obtained using PROFILE and the depletion method in the mass balance
equation (Equation 3) in combination with measured estimates of deposition, leaching and uptake in biomass,
alternative soil balances were estimated (Fig. 4). Since the mass balance method predicted much higher
weathering rates than the other methods, a balance of sources and sinks consequently required more marked
decreases in exchangeable soil stocks for K, Ca and Mg. Furthermore, as a consequence of the substantially higher
$W_{profile}$ for Na, the PROFILE based mass balance suggested substantial increases in exchangeable Na stocks.
**4. Discussion**
**4.1 Comparison of conceptually different methods**
A number of studies have used multiple approaches to estimate weathering rates, with the aim of validating
methods and finding a best estimate for a particular site or catchment (Langan et al., 1995; Kolka et al., 1996;
Sverdrup et al., 1998; Futter et al., 2012). A common problem encountered is that the approaches used are so
conceptually different that the comparisons do not deal with similar quantities at spatial or temporal scale.
Concerning the spatial scale, to our knowledge the mass balance approach has most often been applied at the
catchment and forest stand scale, whereas the depletion method and the PROFILE model are normally applied at
the smaller pedon scale. In the present study, the mass balance approach included data at stand level over a period
when the stand showed high nutrient demand. Concerning the temporal scale, the concepts of the depletion method
and the PROFILE model are conceptually different, although they be applied at a similar spatial scale. These two
methods are based on direct measurements of soil properties, i.e. quantitative mineralogy, soil bulk density and



soil stone content, which is rarely the case in previous comparable studies. Since the three approaches used here
do not measure similar quantities at spatial and temporal scale, and all of them have obvious weaknesses, no
estimate can be taken as a safe reference value of the "true" weathering rate at the study sites. However, for the
purposes of the following discussion, we are of the view that the conceptual differences between these three
approaches are an asset in our case, as they provide complementary information about weathering at different
scales that helps identifying strengths and weaknesses of each method and provide reasons to why these methods
tend to overestimate or underestimate weathering rates of a particular element.
**4.2 Pedon scale weathering rates - a comparison with other studies**
A general finding of this study concerning weathering at the pedon scale was that total modelled  ($W_{profile}$) and
historical ($W_{depletion}$) base cation weathering rates were in good agreement with recent published data for similar
forest sites on podsolic till soils (Stendahl et al., 2013). However, the historical weathering rates at Asa were of
similar magnitude to the lowest historical weathering rate observed by Stendahl et al. (2013). Most Nordic studies
on historical weathering rates have been conducted in the boreal region (Tamm, 1920, 1931; Land et al., 1999;
Olsson and Melkerud, 2000; Stendahl et al., 2013; Starr et al., 2014). Even though soil profile depth and soil age
in these weathering studies differ from those in ours, they obtained higher rates (Land et al., 1999; Olsson and
Melkerud, 2000; Stendahl et al., 2013), similar rates (Stendahl et al., 2013) or lower rates (Starr et al., 2014) for
soils developed on glacial tills.
**4.2.1 Depletion method estimates versus PROFILE model estimates**
A major finding of this study was that, in the 0-50 cm soil profile, $W_{profile}$ was higher than $W_{depletion}$ for all elements
except Na, and the methods generally failed to fulfil our first test criterion concerning weathering in the soil profile
as a whole. However, weathering rates estimated by the depletion method and the PROFILE model can be
expected to differ due to differences in temporal scale. Several studies have concluded that the average historical
weathering rate should generally be higher than the present weathering rate, since soil development involves loss
of easily weatherable minerals and ageing of mineral surfaces (Bain et al., 1993; Taylor and Blum, 1995; White
et al., 1996). In a study using the Historic-SAFE model, applied to the Lake Gårdsjön catchment in southwestern
Sweden, Sverdrup et al. (1998) predicted a decline in weathering rates due to assumed disappearance of fine
particles and loss of minerals. Their results suggested an increase in weathering rates, from the end of the
glaciation 12,000 years B.P. towards a peak at 9000 years B.P., followed by a gradual decrease below initial
levels.
However, in the present study PROFILE generally yielded higher weathering rates than the depletion method at
both study sites. Similar results have been found in other studies, as indicated by high modelled-to-historic
weathering rate ratio ($W_{profile}/W_{depletion}$). At catchment scale, Augustin et al. (2016) found that weathering
estimated by PROFILE was on average 4-fold greater than weathering based on the depletion method. At the
pedon scale, Stendahl et al. (2013) found $W_{profile}/W_{depletion}$ ratios of 2.3 and 2.2 for two sites near Flakaliden,
Vindeln and Svartberget, which are similar values to that found for Flakaliden in our study (2.0). However, the
geographically closest site to Asa (Lammhult) had a much lower ratio (1.1, compared with 3.5 in our study).



Similarly low ratios have been reported for the Gårdsjön site situated in south-western Sweden (i.e. county of Västra Götaland) (Sverdrup et al., 1998; Stendahl et al., 2013), while the high ratio we obtained for Asa was close to that (4.6) reported by Stendahl et al. (2013) for Skånes Värsjö, another site located in south-western Sweden (i.e. county of Skåne). An exception to the general trend of higher steady-state PROFILE weathering rates compared to historical rates calculated by the depletion method, was found for Mg at the Flakaliden site, where $W_{depletion}$ was 1.3-fold greater than $W_{profile}$ in the upper mineral soil, but only at Flakaliden.. This exception with regard to Mg was also found by Stendahl et al. (2013) for all of their 16 study sites.

Our second test, postulating similarity between methods concerning the weathering rate gradient with soil depth, was not fulfilled, because the PROFILE model predicted increasing weathering rates with increasing soil depth, contrary to the depletion method. Since soil-forming processes and ageing of minerals suggest that the present weathering rate might differ from the average historical value, we calculated the hypothetic time needed for the simulated current weathering rates to accomplish the element losses determined with the depletion method. Specifically, one can imagine a front of intense weathering moving down through the soil profile, during which a pristine horizon would undergo an episode, limited in time, of intense weathering followed by slower weathering in the ageing material. In concert with this notion, the highest weathering rate, presently prevailing at approximately 80 cm (Asa) or 60 cm (Flakaliden) depth according to PROFILE (Fig. 2), would cause the observed depletion losses within less than half of the soil age ('max rates' in Fig. 6). However, the calculation also showed that the present minimum weathering rate, simulated for the topmost 1-3 layers (Fig. 2) would often "do the job" in less than the postglacial period ('min rates' in Fig. 6), particularly at Flakaliden, and for K and Na also at Asa. Hence, the current minimum weathering rates according to PROFILE are not generally sustainable within the limits set by the depletion method, indicating either bias in either of the methods, or a current weathering pressure that is correctly modelled but exceeds the historical average. Casetou-Gustafson et al. (*this issue*) could demonstrate that K-feldspar was the dominant source of all steady state PROFILE weathering of K and there are indications that the dissolution rate for K-feldspar is too high compared with mica. For example, Thompson and Ukrainczyk (2002) described differences in the plant availability of K via weathering from these two mineral groups. In addition, Simonsson et al. (2016) found that, although K-feldspar was the dominant K-bearing mineral in a till soil in south-western Sweden, a large proportion of the K loss was explained by mica. Furthermore and in more general terms, Hodson and Langan (1999) suggested that the PROFILE model overestimates weathering rates because it does not consider the decrease in mineral reactivity that has taken place over time and because it assumes that all mineral surface areas are reactive. If this is not accounted for, PROFILE can be expected to generate overestimates rather than underestimates for base cation weathering rates.

As to possible errors in weathering rates according to PROFILE, the sensitivity test indicated that the within-profile variability of soil physical properties had a much greater effect on simulated weathering rates than the differences in mineralogy between different horizons. Therefore, the notion of a weathering front, significantly changing the mineralogical composition of the soil, does not appear to be able to result in any episodic modelled weathering for the studied soils; the decline in weathering rate with time is more likely to be attributable to disappearance of fine particles, as also indicated by the positive correlation between $W_{profile}$ and exposed mineral surface area and bulk density (Figs. S1-S2). Moreover, given the variability of physical and mineralogical





properties in the investigated soils, fine-tuning parameters related to the exposed mineral surface area is most
likely to affect the model output, similarly to the findings of Jönsson et al. (1995). Furthermore, the discrepancy
between $W_{profile}$ and $W_{depletion}$ with respect to soil depth gradient implies that the PROFILE model produces
weathering rate patterns that are not in line with the classic notion of weathering rates being highest in the A- or
E-horizon (Tamm, 1931). However, it is consistent with the more recent notion that mineral dissolution decreases
with increasing time/exposure to weathering (White et al., 1996; Parry et al., 2015). In line with this, the results
support the view that historical weathering rates do not show identical depth gradients to steady-state weathering
rates, and thus $W_{profile}$ and $W_{depletion}$ could both be accurate estimates, of steady-state and historical weathering
rates, respectively.

As to possible bias in the historical weathering rates, underestimates are possible at Asa, where the low values of
$W_{depletion}$ can be attributed to the low gradient of Zr in the soil (Fig. 5). This might, in turn, be the result of soil
mixing by different means. Mechanical soil scarification was carried out at both Asa and Flakaliden prior to
planting of the present stand, which would at least have caused partial mixing or inversion of surficial soil layers.
In addition, clearance cairns of unknown age were found in the experimental area, indicating small-scale
agriculture in the past. Moreover, if burrowing earthworms have been abundant in the past, they might have
produced soil mixing in the upper soil horizons (Taylor et al., 2019), resulting in a complicated and erratic Zr
gradient (Fig. 5) resulting in low estimates of historical weathering in the rooting zone (Whitfield et al. 2011).
High or near-neutral soil pH and deciduous litter can promote high population densities of burrowing earthworms
following forest clearing and agriculture; partly deciduous vegetation indeed dominated at Asa until only 1000-
2000 years BP, with species such as Scots pine (*Pinus sylvestris*), *Corylus avellana* (L.), *Betula* spp., *Quercus*
spp. and *Tilia* spp. (Greisman et al., 2009). Apart from disturbances, natural variability in weathering rates can
likely be attributed to differences in soil texture (i.e. exposed mineral surface area), climate (i.e. temperature and
water percolation rate) and mineralogy. At Flakaliden, the small-scale variation in Zr mobility might be one of
the driving forces behind the large within-site variation in Zr gradient (Fig. 5). The latter is exemplified in different
patterns of Zr enrichment for different soil profiles (Fig. 5). An increase with depth could indicate Zr transport
from shallow to deeper soil layers, most significantly reflected in the Zr/base cation ratio for K (Fig. S3). In
relation to this, in a column experiment Hodson et al. (2002) found that K was the most sensitive element to Zr
mobility and that redistribution of Zr led to insignificant underestimations of base cation weathering. An
alternative explanation for the increased subsoil Zr content could be related to the distinct peaks of rare earth
metals in the B-horizons of Swedish podsolic soils reported by Tyler et al. (2004), who related them to increased
solubility. Increasing Zr concentration in the B-horizon of forest soils in northern Sweden has also been reported
by Melkerud et al. (2000) and, in particular, in a study by Olsson and Melkerud (2000).

Regardless of errors in the Zr gradients, both $W_{depletion}$ and $W_{profile}$ showed more marked gradients with soil depth
at Flakaliden compared to Asa. This could be expected based on the more well-developed podsol profile at
Flakaliden. It has been postulated that the formation of podsols is enhanced by long duration and great depth of
snow cover (Jauhiainen, 1973; Schaetzl and Isard, 1996), which would imply that podsols have developed better
at Flakaliden than at Asa (Lundström et al., 2000). At Flakaliden, the average mass loss of Ca and Mg was 4.0-

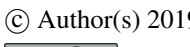



fold larger in the E-horizon than in the B-horizon, which is similar to findings by Olsson and Melkerud (2000) of
a 5-fold higher ratio between losses of base cations in the E- compared with the B-horizon. The average mass loss
since last deglaciation at Flakaliden indicated strong depletion of Ca- and Mg-bearing minerals in the beginning
of the pedogenesis process, which was reflected in the higher historical weathering rates than the steady-state
modelled rates in the upper mineral soil.

The weathering rates of PROFILE may be criticized based on discrepancies in the ranking order of weathering of
elements, compared to historical weathering; this is our third test criterion. PROFILE predicted the highest steady-
state weathering for Na at both sites. However, historical weathering at Asa was greatest for Ca among the base
cation elements, whereas Mg was the most abundant element released at Flakaliden. The latter was also found by
Olsson and Melkerud (2000), who reported the same ranking order of individual base cation weathering (i.e.
Mg>Ca>Na>K) for other sites in northern Sweden.

### 4.3 Weathering in a mass balance perspective

The mass balance approach resulted in consistently much higher weathering rates than PROFILE and the depletion
method for all base cations except Na. This discrepancy was particularly large at Asa. However, it must be stressed
that mass balance estimates of weathering are associated with substantial uncertainties from different sources.
Although the standard error in $W_{MB}$ in the present study was moderate for all elements except Ca, the mass balance
approach most likely underestimated the true uncertainty. Low or moderate standard error was partly due to base
cation uptake in biomass and deposition being calculated as site means only, which would have reduced between-
plot variation in $W_{MB}$ for elements (e.g. K) for which soil changes and leaching were small and accumulation in
biomass was large. Apart from between-plot variation, additional sources of uncertainties originate from
propagation of uncertainties in methods used for each term in the mass balance, e.g. allometric biomass functions,
and precision in soil sampling and chemical analyses. Simonsson et al. (2015) estimated weathering rates for the
Skogaby site in south-western Sweden, a Norway spruce site of similar stand age and soil condition as Asa, using
the same methods as in the present study. Accounting for all sources of uncertainty, they found that the 95%
confidence interval in estimates of base cation weathering was 2.6 times the mean (33 mmol$_c$ m$^{-2}$ yr$^{-1}$). The
confidence interval was the same as the mean value for Mg, but much larger for the other elements.
Despite the considerable uncertainties in $W_{mb}$ estimates, the mass balance approach illustrated that accumulation
in biomass was a dominant sink for all base cation elements except Na. This is in line with findings by Nykvist
(2000) for two Norway spruce sites in Sweden and by Simonsson et al. (2015) for an aggrading Norway spruce
forest in south-western Sweden. However, it contradicts findings in other studies of no change in soil and tree
biomass stocks of base cations (e.g. Sverdrup et al., 1998). The higher estimates of weathering rate at Asa reflected
the higher productivity and nutrient demand of the stand at this site (Bergh et al., 1999), which has resulted in 1.4-
fold greater accumulation of base cations in biomass than at Flakaliden.
The Na fluxes differed from those of the other base cations, probably because Ca, Mg and K are important plant
nutrients whereas Na is not. Calcium and Mg uptake in forest trees is considered to be more or less passive flow

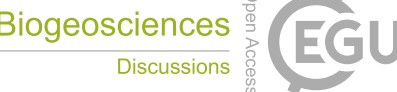

driven by transpiration fluxes, whereas K uptake is an energy-demanding active process (Nieves-Cordones et al., 2014). Considering that Na was the dominant base cation in the soil solution at 50 cm soil depth (Fig. 4), the negligible accumulation of Na in tree biomass suggests that Na uptake in trees is physiologically blocked. Low concentrations of Na seem to be a general feature of terrestrial plants in boreal forests, in contrast to aquatic plants, which explains why the latter are considered important Na sources for large herbivores like moose (Ohlson and Staaland, 2001). Thus, the negligible Na accumulation in tree biomass and the particularly low deposition at Flakaliden simplify the Na mass balance to only three major counterbalancing fluxes: weathering, deposition and leaching. Since $W_{depletion}$ and $W_{mb}$ of Na were fairly similar, and were much lower than $W_{profile}$, our results provide additional support for the claim that the PROFILE model produced consistently too high Na weathering.

Accumulation of Ca, Mg and K in biomass made up the dominant sink. Since deposition and measured depletion of extractable Ca, K and Mg in the soil did not balance this sink, substantial missing sources (here estimated weathering rate) were needed to reach a balance. Using the alternative weathering estimates by PROFILE and the depletion method in the mass balance resulted in even larger estimated depletion in the soil to balance the sinks than was actually measured. Assuming that the measurements of accumulation in biomass, deposition and leaching were reasonably accurate, the results either indicate large uncertainties in measures of soil changes and/or that additional sources of base cations in the soil balanced the sinks. Uncertainties in estimating soil changes were probably significant, since the estimates of soil depletion were based on two single measurements over 12 years and the extraction procedures were not identical over time. Nevertheless, the changes observed in extractable Ca stocks in the soil are in line with observations over 22 years of aggrading Norway spruce forests by Zetterberg et al. (2016), who reported exchangeable Ca depletion rates of 5-11 and 23-39 $mmol_c$ $m^2$ $yr^{-1}$ for sites in south-western and northern Sweden, respectively. The higher value for the northern site reflected higher Ca saturation in the soil. The corresponding values for Asa and Flakaliden were larger, but of similar magnitude (34.5 and 40.5 $mmol_c$ $m^2$ $yr^{-1}$, respectively). Moreover, exchangeable K stocks in the soil normally show little variation over time (B.A. Olsson, unpublished data). Great depletion in exchangeable K stocks in the soil is therefore unlikely. The results therefore suggest that other sources exist in the soil, apart from weathering ($W_{depletion}$, $W_{profile}$) and depletion of ammonium-chloride-extractable base cation stocks. It is well known that the exchangeable nutrient stock in the soil is defined by the extraction medium and procedure. A test of different extractants used on the soils in the present study revealed that using $NH_4OAc$ posed a risk of underestimating the amounts of base cations in the soil and that the yield of exchangeable base cations decreased in the order aqua regia > HCl > EDTA > $BaCl_2$ > $NH_4OAc$ (Olofsson, 2016). Using a more potent extractant than 1 M $NH_4Cl$ would probably have resulted in different findings on the change in extractable and plant-available base cations in the soil. Regarding K, fixed or structural K in clay minerals provides a dynamic pool of K that is not included in modelled weathering or in $NH_4Cl$-extractable K (Simonsson et al., 2016). Regarding Ca and Mg, dissolution from non-crystalline/amorphous compounds can be an important source in soils depleted of these elements (Van der Heijden et al., 2017).

Another possible explanation for the higher weathering rates with the mass balance approach is that PROFILE may produce conservative estimates of weathering because the model only captures steady-state chemical processes. It has been postulated that e.g. mycorrhizae play an important role in nutrient uptake in forest trees



through active foraging by mycelia at mineral surfaces, but the nature and potential importance of biotic control
of weathering has been much debated in recent decades (Finlay et al., 2008; Sverdrup, 2009; Smits and Wallander,
2017). Based on the results of the present study, the hypothesis of significant biological control of weathering was
not rejected.
**5. Conclusions**
The depletion method, PROFILE model and mass balance approach was used to estimate weathering rates at two
coniferous forest sites in Sweden. The methods estimated weathering rates at different spatial and temporal scale,
and no estimate was taken as a reference value of the true (current) weathering rate. There was no similarity in
weathering estimates between the depletion method and the PROFILE model with respect to BC weathering in
the 0–50 cm soil layer and the soil depth gradient in weathering rates except that both methods indicated higher
weathering rates and more marked depth gradients at Flakaliden compared to Asa. The PROFILE model produced
consistently higher weathering rates than the depletion method except for Mg, and while the depletion method
estimated decreasing weathering rates with increasing soil depth, the PROFILE model predicted the opposite. The
mass balance method produced significantly higher weathering rates for all elements except Na. A cross-
examination of the estimates stressed the importance of that all criteria for application of the depletion method
must be satisfied. Erratic or weakly developed Zr gradients in the soil, possibly caused by natural and
anthropogenic disturbances can be a cause to why the depletion method underestimates weathering rates. The
higher weathering rates of K by PROFILE compared to the depletion method could be an indication of that
inaccurate dissolution rates of K-bearing minerals was used in the model. On the other hand, high mass balance
estimates for K, Ca and Mg weathering suggests that there were additional sources of base cations for tree uptake
in the soil besides weathering and measured depletion in exchangeable base cations, and that PROFILE produced
conservative estimates of base cation supply to forest trees.
**6. Authors contribution**
Authors contributed to the study as in the following: S. Casetou-Gustafson: study design, data treatment, analyses,
interpretation and writing. Magnus Simonsson: study design, analysis, interpretation and writing. Johan Stendahl:
study design, analysis, interpretation and writing B.A. Olsson: study design, data treatment, analysis,
interpretation and writing. S. Hillier: interpretation and writing. Sune Linder: Provided long-term experimental
data, interpretation and writing. Herald Grip: Provided long-term experimental data, interpretation, and writing
**7. Acknowledgements**
Financial support from The Swedish research Council for Environment, Agricultural Sciences and Spatial Planning
(212-2011-1691) (FORMAS) (Strong Research Environment, QWARTS) and the Swedish Energy Agency
(p36151-1). Stephen Hillier acknowledges support of the Scottish Government's Rural and Environment Science
and Analytical Services Division (RESAS). We thank Cecilia Akselsson for her contribution to study design,
PROFILE model development, interpretation and writing.



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



**Table 1.** Soil profile characteristics at 50 cm depth in the mineral soil at the Asa and Flakaliden sites

| Site | Plot | Clay (%wt) | Silt (%wt) | Sand (%wt) | Coarse (%wt) | Density (g/cm³) | Soil age (calendar years) |
|------|------|-----------|-----------|-----------|-----------|-----------|-----------|
| Asa | K1 | 9.49 | 25.04 | 45.30 | 20.18 | 1.10 | 14300 |
| | K4 | 7.65 | 22.59 | 39.21 | 30.48 | 1.09 | 14300 |
| | F3 | 4.95 | 25.26 | 40.54 | 29.25 | 0.99 | 14300 |
| | F4 | 8.64 | 25.69 | 40.13 | 25.54 | 0.94 | 14300 |
| Flakaliden | 15A | 1.92 | 9.21 | 68.98 | 19.68 | 1.89 | 10150 |
| | 14B | 7.71 | 34.09 | 33.71 | 24.17 | 1.35 | 10150 |
| | 10B | 7.75 | 45.17 | 37.23 | 8.90 | 1.36 | 10150 |
| | 11B | 9.56 | 45.07 | 33.91 | 10.72 | 1.47 | 10150 |



**Table 2.** Concentrations of different elements at the reference depths used for calculating historical weathering rate at the Asa and Flakaliden sites

| Site | Plot | Ref. depth (cm) | Ca (%) | Mg(%) | K(%) | Na (%) | Zr (%) | Ti (%) |
|------|------|-----------------|--------|-------|------|--------|--------|--------|
| Asa | K1 | 80-90 | 1.41 | 0.51 | 0.93 | 1.06 | | 0.34 |
| | K4 | 80-90 | 1.29 | 0.44 | 0.88 | 1.00 | 0.00029 | - |
| | F3 | 60-70 | 1.41 | 0.55 | 0.87 | 1.04 | 0.00028 | - |
| | F4 | 80-90 | 1.26 | 0.49 | 0.85 | 0.98 | 0.00029 | - |
| Flakaliden | 10B | 60-70 | 1.09 | 0.57 | 0.88 | 0.87 | 0.00024 | - |
| | 11B | 50-60 | 1.63 | 0.79 | 0.82 | 1.06 | 0.00043 | - |
| | 14B | 60-70 | 1.59 | 0.70 | 0.81 | 1.03 | 0.00034 | - |
| | 15A | 60-70 | 1.46 | 0.59 | 0.94 | 1.15 | 0.00025 | - |















**Table 3.** Description of parameters used in the PROFILE model

| Parameter | Description | Unit | Source |
|---|---|---|---|
| Temperature | Site | °C | Measurements at Asa and Flakaliden |
| Precipitation | Site | m yr$^{-1}$ | Measurements at Asa and Flakaliden |
| Total deposition | Site | mmol$_c$ m$^{-2}$ yr$^{-1}$ | Measured data on open field and throughfall deposition available from nearby Swedish ICP Integrated Monitoring Sites |
| Base cation net uptake | Site | mmol$_c$ m$^{-2}$ yr$^{-1}$ | Previously measured data for Asa and Flakaliden: Concentrations in biomass from Linder (unpublished data). Biomass data from Heureka simulations. |
| Net nitrogen uptake | Site | mmol$_c$ m$^{-2}$ yr$^{-1}$ | Previously measured data from Asa and Flakaliden: Concentrations in biomass from Linder (unpublished data). Biomass data from Heureka simulations. |
| Base cations in litterfall | Site | mmol$_c$ m$^{-2}$ yr$^{-1}$ | Literature data from Hellsten et al. (2013) |
| Nitrogen in litterfall | Site | mmol$_c$ m$^{-2}$ yr$^{-1}$ | Literature data from Hellsten et al. (2013) |
| Evapofraction | Site | Fraction | Precipitation data and runoff data. Runoff data been calculated based on proportion of runoff to precipitation (R/P) at Gamttratten and Anneboda. |
| Exposed mineral surface area | Soil | m$^2$ m$^{-3}$ | Own measurements used together with Eq. 5.13 in Warfvinge and Sverdrup (1995) |
| Soil bulk density | Soil | kg m$^{-3}$ | Own measurements |
| Soil moisture | Soil | m$^3$ m$^{-3}$ | Based on paragraph 5.9.5 in Warfvinge and Sverdrup (1995) |
| Mineral composition | Soil | Weight fraction | Own measurements |
| Dissolved organic carbon | Soil | mg L$^{-1}$ | Previously measured data for Asa and Flakaliden: Measurements for B-horizon from Harald Grip and previously measured data from Fröberg et al. (2013) |
| Aluminium solubility coefficient | Soil | kmol m$^{-3}$ | Own measurements for total organic carbon and oxalate-extractable Al together with function developed from previously published data (Simonsson and Berggren, 1998) |
| Soil solution CO$_2$ partial pressure | Soil | atm. | Based on paragraph 5.10.2 in Warfvinge and Sverdrup (1995) |







**Table 4.** Assessment of data quality for terms included in the mass balance estimate of weathering

| Term | Spatial scale | Temporal scale | Data source | Quality of term quantification |
|---|---|---|---|---|
| Deposition | Adjacent sites | Annual or monthly measurements | Svartberget experimental forest, and Integrated Monitoring site | Moderate: high quality of data, but estimates are not site-specific |
| Soil stock change | Site (initial) and plot (repeated) | Repeated samplings (4) | Unpublished data from J. Bergholm and H. Grip. Olofsson (2016) | Moderate/low: repeated sampling biased by differences in methods of sampling and soil extraction. |
| Leaching | Plot | Sampling of soil water at 50 cm depth repeated 3 times per year. Water flux modelled (COUP). | H. Grip, unpublished data | High/moderate: High spatial and temporal resolution in soil chemistry, but uncertainty in separating lateral and vertical flow (Flakaliden). |
| Biomass accumulation | Site (control plots) | Growth increment measured from biomass studies at start and after 12 years. | Growth Albaugh et al. (2009) Nutrient content: S: Linder unpublished data | High/moderate: High quality in growth estimates and nutrient content at treatment scale, data lacking at plot scale |














**Figure captions**

**Figure 1.** Titanium (Ti) to zirconium (Zr) ratio (by concentration) used as an indicator of uniform parent material in all soil layers at Asa (F3, F4, K1, K4) and Flakaliden (10B, 11B, 14B, 15A).

**Figure 2**. (Left) Historical weathering rate of base cations (mmol$_c$ m$^{-2}$ yr$^{-1}$) estimated by the depletion method and (right) steady-state weathering rate estimated by the PROFILE model in different soil layers at Asa and Flakaliden.

**Figure 3.** Comparison of weathering rates (mmol$_c$ m$^{-2}$ yr$^{-1}$) for Ca, Mg, Na and K determined with the depletion method, the PROFILE model and the mass balance method for the 0-50 cm layer at Asa and Flakaliden. Each error bar represents the standard error calculated based on four soil profiles at each study site.

**Figure 4.** (Left) Sources and (right) sinks of base cations in ecosystem net fluxes at Asa and Flakaliden. The soil is a net source if soil base cation stocks decrease and a net sink if they increase. 'Mass balance' = current base cation weathering rate (W$_{BC}$) estimated with the mass balance method, including measured changes in soil extractable base cation stocks; 'PROFILE' = soil extractable pools estimated from mass balance using PROFILE estimates of steady-state weathering rate; 'Historical' = soil extractable pools estimated from mass balance using estimates of historical weathering rate by the depletion method. 'Measured soil change' and 'Mass balance estimated soil change' indicates that equation 3 was used to estimate weathering rate or the soil change, respectively.

**Figure 5**. Zirconium (Zr) gradient in the soil at Asa (KI, K4, F3, F4) and Flakaliden (10B, 11B, 14B, 15A)

**Figure 6.** Time (years) required to achieve the measured historical element loss in different soil layers with maximum or minimum PROFILE weathering rates at (a) Flakaliden and (b) Asa.





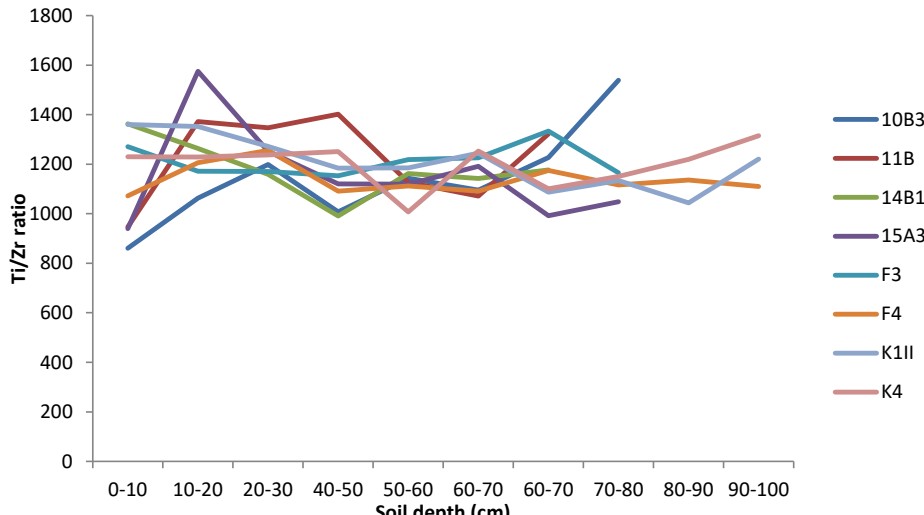


**Figure 1.**







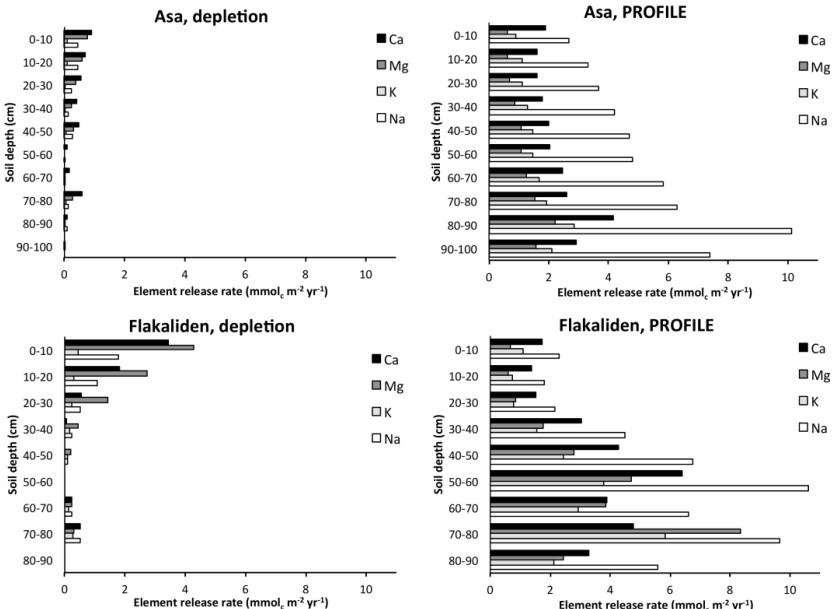


**Figure 2.**









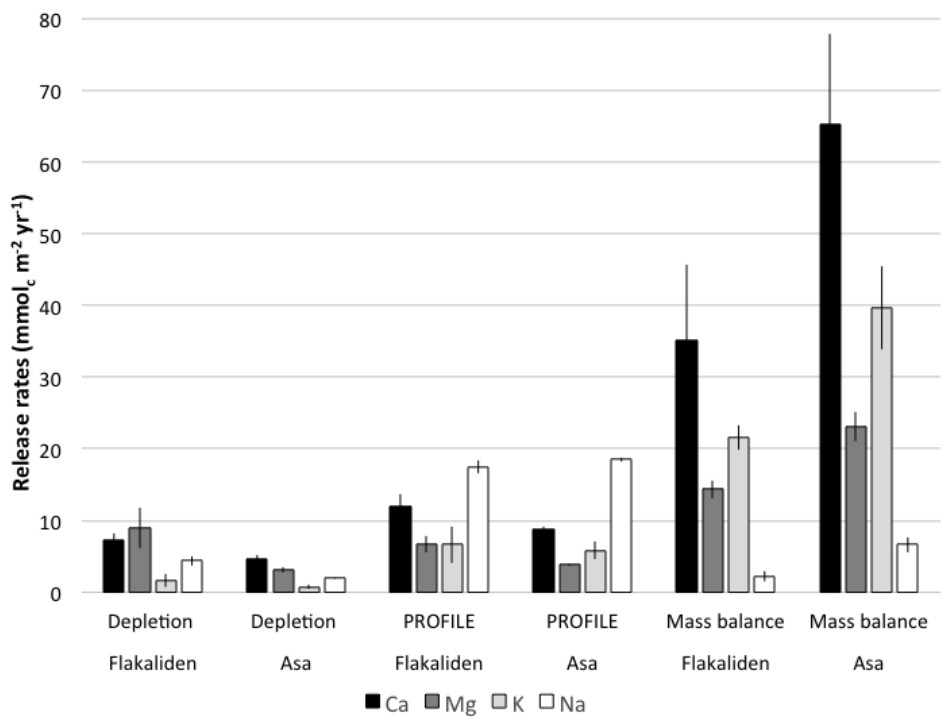

**Figure 3.**





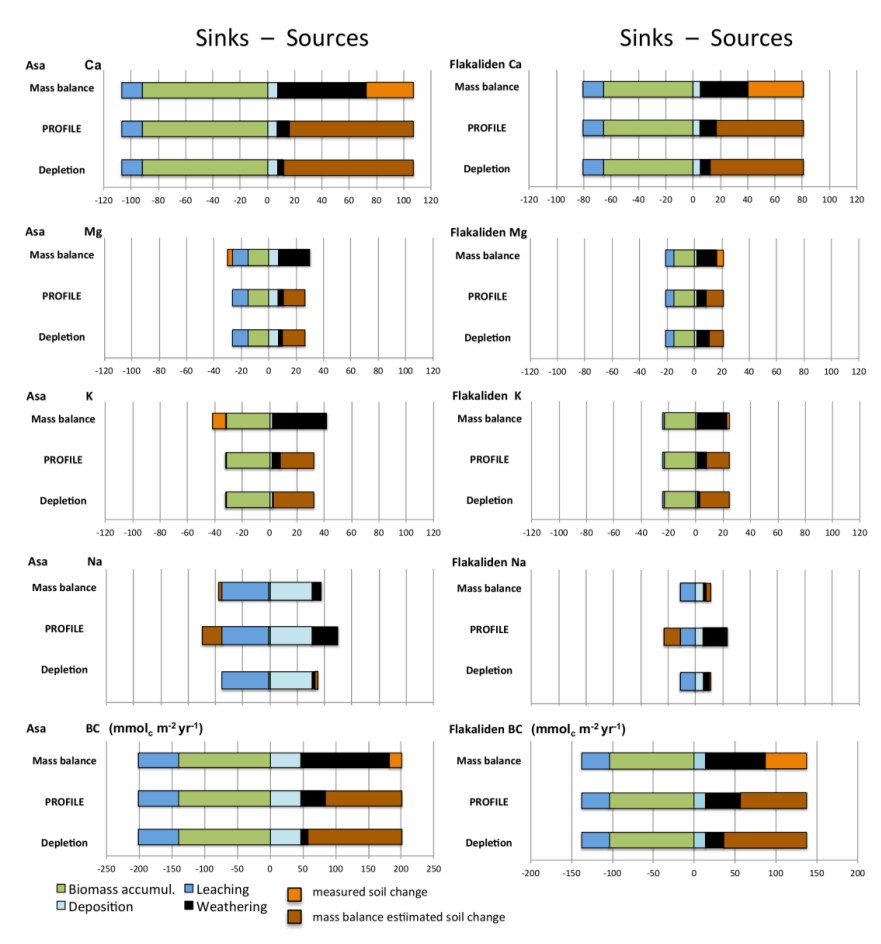


**Figure 4.**



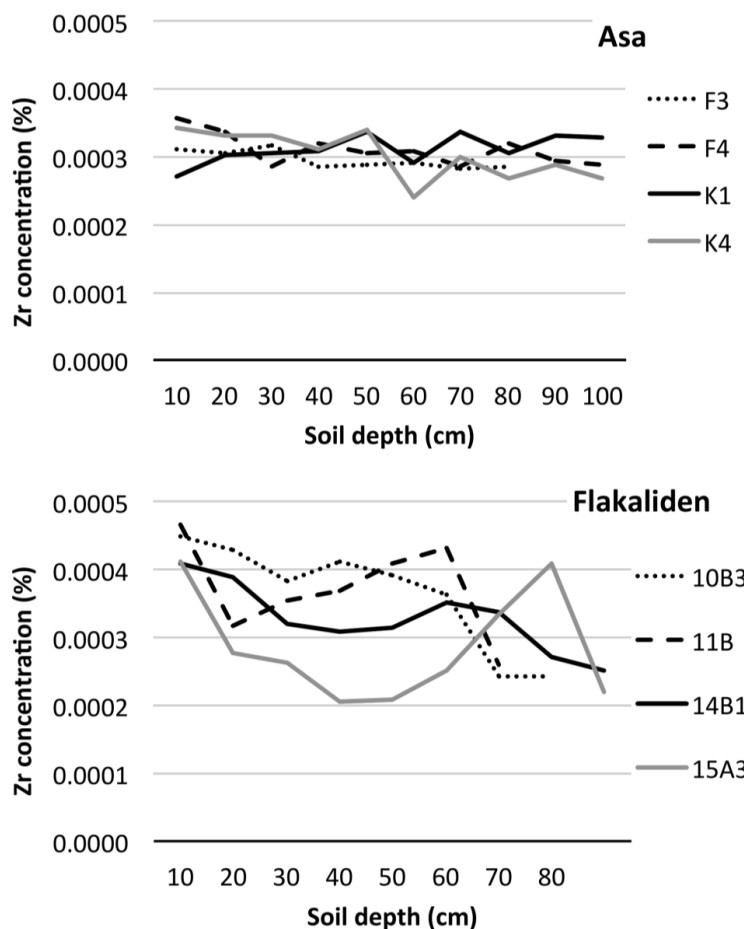


1126        **Figure 5**.





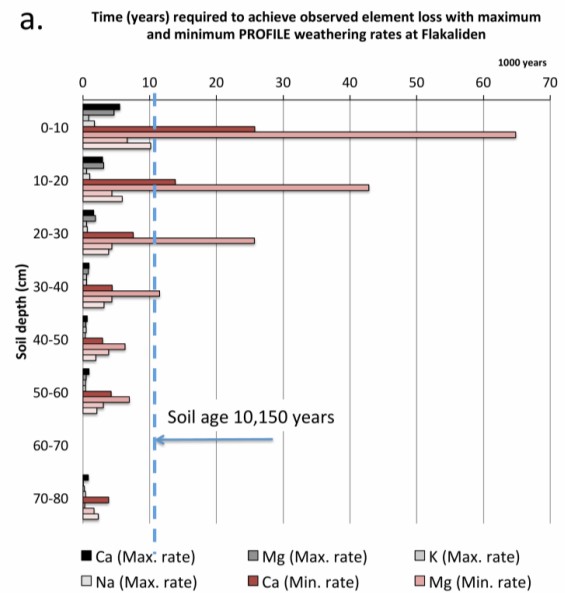


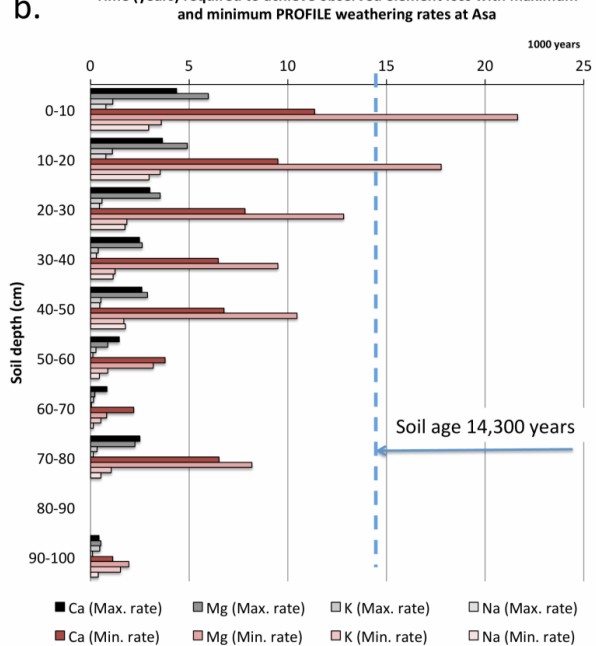

1129        **Figure 6.**