# Peer review of "Current, steady-state and historical weathering rates of base"

_Biogeosciences, 2019_

## Referee Comment (RC1) · Anonymous Referee #1 · 28 Feb 2019

This paper compares three methods (elemental depletion, PROFILE and catchment budget) for estimating base cation weathering rates at two forested sites in Sweden. I understand that this is part of a special issue, but in my opinion, this paper offers nothing new to the literature except that these established methods have been applied to two new sites. Given the lack of any originality, limited sampling and an incomplete consideration of the large uncertainties associated with all these methods I do not feel that it should not be published. There must be at least 10 previously published papers (some of which are cited; e.g. Klaminder, Futter, Bain, Whitfield etc.) that have done

the same thing as presented here and in general they all show the same thing – base cation catchment budgets are typically the largest (as some base cations are lost from the exchange pool), followed by PROFILE with the elemental depletion method usually being the lowest. The fact is that all methods have large uncertainty, and this is not adequately captured in this paper. For example, we know for sure that the budget model cannot adequately capture weathering as there are always losses/gains from the exchangeable pool and soils are heterogeneous. The concluding sentence in the abstract "The large discrepancy in weathering rates for Ca, Mg and K between mass balance and the other methods suggest that there were additional sources for tree uptake in the soil besides weathering and measured depletion in exchangeable base cation" is completely unfounded and should be removed. The inaccuracies in all the methods are such that the discrepancy is most likely because of uncertain weathering rates combined with unknown variability associated with changes in soil exchangeable pools and uptake. Further, in the abstract mean numbers are presented, but each of these has a large error associated with it so what is the point? The study design itself is such that the data are extremely uncertain – four plots at each site and a single sample from each plot? [4 samples per site]. This is insufficient in my opinion. As stated – the comparison between PROFILE and elemental depletion has been done probably at least 10 times before.

The introduction and methods that describes these methods in detail is essentially a repeat of other studies – I have read this before. The PROFILE uncertainty appears to be part of another paper in this special issue which further questions why this paper should be published. The authors show that surface area affects weathering estimates a lot in PROFILE (which we know), but it does not seem that the authors actually use mineral surface area but use bulk soil surface area, which is not the same thing. Other inputs to PFOFILE are estimated and ultimately lead to considerable uncertainty in the estimates.

The mass balance is uncertain because 1) soil is so variable meaningful changes cannot be determined accurately [probably the most uncertain part of this budget], 2) forest growth was assessed at the plot level (not at the pit level) and tree chemistry in addition to biomass is very uncertain/variable, 3) export is essentially modelled [runoff] and sampling frequency is limited. Simply judging the parameters as high or moderate quality (lines 405-407) is a subjective aspect that should be included in a scientific manuscript. Given the fact that really this paper provides no knowledge beyond describing the data that the authors have for these two sites there is little point commenting on the paper in general. It is far too long, compares mean values estimated by different weathering rates [which is pointless for reasons mentioned above and also because in some case samples were "excluded"]. I don't believe that this paper does anything that previous papers did so while they have used approaches that others have used – they of course get the same results, but at two different sites. I don't think that this is sufficient to warrant publication, but I expect that as it is part of a special issue it may be decided to keep it. Given the lack of novelty and my opinion that it should not be published I do not wish to make a lot of editorial comments as my job is to decide whether the paper is a novel contribution or has major flaws (see above), not to edit a paper that I do not think meets these criteria.

---

## Referee Comment (RC2) · Anonymous Referee #2 · 12 Mar 2019

The manuscript utilized three different methods and quantified weathering rates in two forested sites in Sweden. This study is very interesting, and I applause such efforts, as most studies focused on just one method. So this manuscript may be of interest to the community that Biogeosciences serves. However the manuscript, as currently written, made it difficult to understand exactly how three methods work, and evaluate the uncertainties from each method, in order to compare them for magnitude, depth gradients etc of such weathering rates, and provide recommendations how these methods are used for future studies. Below I provided ways to improve, followed by detailed

[Figure]

comments.

(1) The title of the manuscript suggests that this study focuses on the comparison of three methods and examines how they are similar or different in quantifying the weathering rates. If so, the details of these three methods need to be explained in great details.

Historical weathering rates This is also called "long-term" weathering rate in the literature, and is estimated from soil chemistry/mineralogy. A few important considerations are missing from the paper.

The balance of chemical degradation versus physical erosion. What are the topographic features in the study sites? If the areas are flat and minimum physical erosion is present, this needs to be stated clearly in the paper. In contrast, if the erosion loss is important, then the top layer that was most depleted had been eroded away and the historical weathering rates will be underestimated.

The equations used need to be included in the paper, in addition to the citations (Marshall and Haseman (1943) and Brimhall et al. (1991)). For example, bulk density data were used as input data, but the method did not specify how BD was used. I am specifically interested in mass transfer coefficients and the volume change factors. For example, if the soil collapsed or expanded during chemical weathering, then the sample interval 10-20 cm, for example, may not be from 10-20 cm of original sediment before collapse or expansion. If so, the gradients or the total weathering rates may not be represented well. More importantly, the overall weathering rates, plotted in Figure 3, will be off.

QA/QC is needed for elemental data, including Zr, as this is important to evaluate the mobility of base cations as chemical weathering occurs. Zr concentration is extremely low as reported in Table 2, around 2 to 5 ppm. What is the precision in these low concentrations?

Line 254-Line 257 and Line261-Line262: How is the soil/sediment from a certain depth chosen as parent for a weathering profile needs more discussion. The manuscript assumed constant Zr concentrations and explored this by plotting Zr/Ti ratios in Figure 1. This is okay, but what about the homogeneous nature of these glacial sediments, in terms of base cations? For example, does major elemental chemistry remain constant after the 60-70 cm? This is critical to evaluate the error bars for mass transfer coefficients and weathering rates. By looking at data from Table 2, parent chemistry is different among all sites. Does that imply glacial sediments have varible chemistry with depth too?

Age: time used for quantifying historical weathering rates is the age of the landscape, i.e., the time since glacial retreat. However this is the age of the soil at the surface, and soils at deeper depths should be much younger. Indeed, as weathering proceeds, soil thickness increases ($\sim$70 cm according to this manuscript in southern and northern Sweden). If so, the age of soils decreases with depth. At the soil/parent boundary, the soil age should be zero. If so, how can release rates of cations be quantified using one age? This probably also explains partially why weathering rates are much higher at surface than at depth in Figure 2.

PROFILE model I understand that several other manuscripts are under review on the PROFILE model in the same issue. However since this manuscript stands on its own, and the PROFILE model is compared to other methods. It is reasonable to provide enough details for PROFILE model in this manuscript too.

What is a steady state weathering rate? Can you define it and also justify why steady-state is assumed? The introduction mentioned S emission and forest acidification etc. Would this allow a steady-state status at this?

Some default values, including pCO2, soil moisture etc are used for PROFILE modelling. Can you make a table to show how these variables are different or similar among all sites? This is very important as you start to compare sites for different weathering

rates in Results and Discussion.

For sensitivity analysis, in section 2.4.3, three groups of variables are listed. Can you list values or ranges of these variables in the table above? A homogeneous soil profile is used. Can you write a couple of sentences to explain why such a scenario is chosen? I am also interested in the residence time of soil water at each depth interval.

Base cation mass balance I think it is critical to explain how much uncertainty is induced by different assumptions. For example, in quantifying the uptake in biomass (BBC), mass of fine roots was assumed and below ground biomass were estimated. How much error is this in the overall BBC?

The canopy budget method was used to quantify the inputs from atmospheric deposition (introduced between Line 304-Line 310). However, I am not sure I understand how it is done. Can you elaborate? How is dry deposition data used to estimate "dissolved" load of base cations?

Leaching (LBC) is estimated using three soil water samples collected at 50 cm. Is this for one year only, or multiple years? This is not an optimum method as soil water chemistry is known to vary significantly over seasons (e.g., a function of soil water residence time, biological uptake etc). Can you explain how different water chemistry is and when exactly the samples were collected? The evapotranspiration is estimated by CoupModel. This is entirely model-based, and is there any way to validate this?

Does plagioclase or amphibole weather to form smectite or other clay minerals and thus retain the cations?

Table 4 is a great effort to assess the data quality and evaluate the uncertainties in quantifying WBC using mass balance approach. Some are already discussed in the Method section, in terms of data source and spatial/temporal scales of data collection, and the only column added is the quality of term quantification. However the term quantification is so qualitative, and what is the cumulative errors in WBC? The leaching is listed as high temporal resolution. I do not think 3 sampling over one year is considered high temporal resolution though.

(2) Discussion Line 521-Line 529: I am not sure I understand this discussion. The weathering rates at pedon scales are compared to others. Should they be similar or different and Why? These statements are not useful if they are not put into some context where sites can be compared, in terms of soil age, types of parents, etc.

Line 542-555: similarly, this discussion cited several studies on different Wprofile/Wdepletion ratios, but did not specify where these ratios are above or below 1. If so, what do we learn?

Line 586-Line 587: The correlation is observed between weathering rates and bulk density. What does this mean? How does bulk density change the mineral dissolution kinetics? Why is it linear? Also according to the method, bulk density is estimated and it increases with depth. If so, is this correlation between weathering rates and bulk density an artifact?

Line 598-Line 620: It is frustrating that the depletion method was introduced and weathering rates were calculated and discussed, and then here the authors explained that Zr may not work as an immobile element, and the depletion method is not working.

(3) In Figure 3, weathering rates by three methods for two sites are compared. Error bars are added to represent the uncertainties for each rate. However it is not explained exactly how error bar is derived. This is very important.

Minor comments: Acid silicate bedrock: I have never seen this phrase before. Do you mean siliceous or felsic? Or do you really mean the bedrock is low in pH?

Line 94: recovery is driven mainly by silicate rocks? What about carbonate rocks? Line 181: different font used here. Line 183: Missing "rates" after weathering. Line 197-Line 200: What nutrients were added as solid fertilizers? Are there base cations added as well? This is important to know for mass balance calculation. Line 216: Use either 143

000 and 10 150 years, or 143 and 10 thousand years. Line 255: space between "and" and "above". Line 267: Figure 5 is cited before Figures 2-4? Can you renumber them? Line 425: Misspelling: Weatherable? Avoid paragraphs with just one sentence, such as those in Line 487-Line 489.

―――――――――――――――――――――

---

## Referee Comment (RC3) · Anonymous Referee #3 · 13 Mar 2019

The evaluated paper focused on detailed evaluation of chemical weathering rates of base cations in two representative sites of Sweden. It is very important task for Swedish forestry management strategies. There is a debate which form of tree removal during harvest should be preferred. Tree harvest with stem only is probably better than the newer method of whole-tree harvest at least with respect of nutrient base cations: calcium, magnesium and potassium. The presented research is not just an academic comparison of the following three assessments of weathering rates 1) elemental depletion method in soil representing mostly historical weathering rates 2)

[Figure]

PROFILE modelling of soil representing mostly steady-state weathering rates 3) mass balance method in the pedon scale representing mostly current weathering rates (NOT mass balance in the CATCHMENT scale as mentioned incorrectly by the Reviewer 1). The paper is well written and very interesting. Presentation of main results is straightforward and valuable, in the main text and in some figures as well (e.g. Figs. 3 and 4). I recommend publishing this manuscript as soon as possible. It could be published alone but it will have much higher impact to be published in the special issue of Biogeosciences dedicated to the assessments of chemical weathering rates for sustainable forestry.

I recommend different orientation of Fig. 1 and Fig. 5, with depth in the vertical axes (like in the Figs. 2 and 6). Unfortunately I did not find captions for Supplementary data.

Short comments: 153 (5) – should be Erlandsson Lampa (not Lampa Erlandsson); 181 (6) – improve one sentence (. . ..on soil derived from glacial till soil derived from mostly acid. . ..); 255 (8) – . . ..reference layer andabove the reference layer. ???; 305 (10) – Please add the distance from your research sites to the IM catchments (Aneboda and Gammtratten); 307 (10) – I do not think that sodium is prone to canopy leaching (and perhaps also sulphate); 397 (12) – I am surprised that the soil water was collected only 2-3 times per year; 425 (13) – weatherable, not weatehrbale; 554 (17) – one redundant ".”; 870 (25) – should be Erlandsson Lampa (not Lampa Erlandsson); 955 (27) – 1990– 1999 (not ?); Tab. 1 (30) – soil age should be probably 143000 (like in the text 216), not just 14300 years at Asa; Tab. 2 (30) – you should add "extractable" because some readers could think that the concentrations of base cations are "exchangeable"; Tab. 3 (31) – should be Aneboda (not Anneboda); Fig. 4 (37) – I recommend to delete site names from all 10 figures and write them only below Sinks – Sources title; Fig. S1 – add the right parenthesis to all four figures

---

## Author Comment (AC1) · 4 Apr 2019

Answer to referee 2:

We would like to thank the referee for helpful comments. (1) We agree with the referee and think it would be possible to give more detailed method description. - Physical erosion: The effect of soil erosion is negligible at our sites due to relatively flat terrain, dense ground vegetation cover, stable till soil parent material and young soil age. The soil surface has been covered by dwarf-shrubs, lichen and mosses for very long time,

resulting in low erosion. A general evidence for long time vegetation cover is the rapid vegetation establishment in areas where glaciers are retarding. (2) -Equation for calculation of weathered loss of mobile elements: We agree with the referee and will include an equation developed by Olsson and Melkerud (1989), which is based on the same principles as the mass transfer function. -Volume change factors: Fractional volume change (Vp) according to e.g. White et al. (1996) was calculated in Stendahl et al. (2013) for similar soil types (not presented in paper) showing positive values down to 40 cm depth indicating volumetric expansion, and further below it was 0 indicating no volume change. Data on Vp could be added to the current manuscript as well. -Table 2: By mistake we reported Zr in the wrong unit as well as showing too many decimal places after the comma. Therefore, we agree with the referee, we should report Zr in ppm. -Assumption about homogeneous parent material: The BC/Zr-ratio (Figure S3) was used to help defining the reference soil layer as it shows heterogeneities in parent material with depth. In case of heterogeneities in the profile the reference layer was chosen above this heterogeneity. -In response to the explanations given by the referee about depth gradients (i.e. "However this is the age of the soil at the surface, and soils at deeper depths should be much younger. Indeed, as weathering proceeds, soil thickness increases (âĹij70 cm according to this manuscript in southern and northern Sweden). If so, the age of soils decreases with depth. At the soil/parent boundary, the soil age should be zero. If so, how can release rates of cations be quantified using one age? This probably also explains partially why weathering rates are much higher at surface than at depth "), we argue: Our ultimate purpose is to estimate the long term average weathering per m2 per year. In our calculations of mean annual depletion since deglaciation, 'age' is simply a constant number of years at each site. The degree of change in a soil layer should then be referred to as 'loss' or 'change', not 'age'. Of course this means that we (and many others using the depletion method) assume that weathering has been more or less constant over time, which is not quite true since there is a gradual decline in weathering rate due to loss of minerals, formation of coatings etc. However, in these very young soils the assumption is probably more accurate.
(3) -Steady-state description: Steady-state means that the state is dependent on the ecosystem properties and mechanisms that prevail under current climatic and biotic influence (i.e. PROFILE simulates present day weathering rates under steady conditions, which means that all parameters are kept on a constant level). The PROFILE model was developed in the context of determining criritcal loads of aciditiy, where the major interest were long-term sources and sinks of acidity (i.e. processes that are irreversible over the time scale of decades). As such, Sverdrup et al. (1990) justifies the use of long-term kinetics (i.e. steady-state kinetics) for mineral dissolution for young soils (such as Swedish soils). -Parameter table: It is possible to describe the input parameters in a simplified way in a table (i.e. not per soil layer, bur per site or alternatively showing ranges). An example of an input parameter table is given in Akselsson et al. (2004, 2016). - Reporting data used in sensitivity analysis: We have reported these data in the Supplementary material (Table S1 and S2) - Residence time of soil water: PROFILE is not a dynamic model. Parameters are not modeled as a function of time. The weathering rate for a specific soil layer is dependent on the degree of soil moisture saturation. Weathering reactions can only take place on wetted surfaces and PROFILE assumes that all exposed mineral surfaces ares are wetted. Precipitation input data are used to describe the vertical water flux through the soil profile. The flux leaving a soil layer is calculated based on annual average precipitation and runoff in relation to the relative base cation uptake. Both the water flux and the soil moisture content are fundamentally involved in calculating soil solution equilibria for each soil layer based on steady state mass balances (Equation 3.2 to 3.6 in Warfvinge and Sverdrup, 1995).

(4) The referee asked: "How much error is this in the overall BBC"? We answer: The revised version will include estimated uncertainty ranges (equivalent to 'confidence intervals') for the base cation budgets and weathering derived from them, based on information on between-plot variability and, where this is not available from the present experiments, on estimated uncertainties calculated by Simonsson et al. (2015; Table 4-8 in that paper); the field trial of that paper has large similarities to the present studies regarding the study design.

- Total atmospheric deposition: We agree with the referee. It is not necessary to refer to the canopy budget model, since canopy exchange is not considered by PROFILE. However, by calculation of dry deposition factor we correct for canopy exchange. In order to distinguish canopy exchange form dry deposition for Ca, Mg and K, Na was used as a tracer ion. Dry deposition for Na and Cl was calculated as the difference between wet and throughfall deposition. Wet deposition for all elements was calculated as outlined by Zetterberg et al. (2014), i.e. correcting bulk deposition for dry deposition using wet-only to bulk deposition ratio. Finally, total deposition for all elements was estimated by summing up dry deposition and wet deposition. -With regard to the leaching, the referee wrote:" Leaching (LBC) is estimated using three soil water samples collected at 50 cm. Is this for one year only, or multiple years? This is not an optimum method as soil water chemistry is known to vary significantly over seasons (e.g., a function of soil water residence time, biological uptake etc). Can you explain how different water chemistry is and when exactly the samples were collected?" And we argue: Soil water sampling was performed twice every year, i.e. in the spring and in the autumn, which is the period of highest water flux and which means that the most important leaching events are covered. An additional illustration of the variation in water chemistry with time of the year could be given as supplementary figure to show BC fluxes for this period and to compare with the drainage flux: Mean and Standard error of BC ions (mg/l) in soil water sampled at 50 cm depth in the soil of the four control plots at Asa during 1990 – 2004. The samples were collected twice every year, in the spring and in the fall. The spring samples were collected soon after the snowmelt and depending on current weather a specific year this meant that the yearly spring sampling date varied between the last week of April and the last week of May. The fall samples were collected late, i.e. soon before frost in the soil was expected. That meant that the fall sampling dates varied from year to year, i.e. from the first week in September to mid-November. - Evapotranspiration: We agree with the referee, it is entirely model-based. However, there are ways to validate it, i.e. Alternative ways to estimate water balance can be used. Calculated drainage compare well with nearby catchment runoff and drainage

and evapotranspiration compare well with Swedish Meteorological and Hydrological Institute (SMHI) results from our study regions. -The referee asked: "Does plagioclase or amphibole weather to form smectite or other clay minerals and thus retain the cations"? We answer: Formation of smectite, or any other clay mineral with a CEC, potentially increases the pool of exchangeable cations in the soil. Small amounts of smectites are known to form in podzolized soils (Olsson and Melkerud, 1989). However, at the time scale of the present cation budget, which involved a fraction of a tree rotation (a few decades), we can expect changes associated with cation accumulation in the biomass to vastly outnumber any changes in CEC due to any clay formation. To the extent it occurs, any increase in exchangeable cations due to clay formation is included in the base cation budget; therefore, it does not represent a methodological problem. By contrast, release and fixation of non-exchangeable cations in clay minerals, e.g., of interlayer K (which may be significant even in soils poor in clay minerals; Simonsson et al., 2016), contribute positively or negatively, respectively, to the unknown sources and sinks summarized as the 'weathering'. - Discussion Line 521-529: We agree with the referee that this section can be improved and we suggest that we merge 4.2 and 4.21 and keep the title from 4.2.1 and then we decide what parts of 4.2 are still relevant. We should also introduce a sentence directly in the beginning of 4.2 "Our first test criterion was to see if….". The test criterion can be estimated from the magnitude of weathering but also from the proportions of the estimates of depletion versus PROFILE." When we speak about the magnitude of weathering, maybe we should focus on soils of similar pedogenesis (Swedish studies) and where the same soil depth has been considered for weathering calculations. We should then write about high similarities or/and mention studies where there are dissimilarities. -Discussion Line 542-555: The discussion about different $W_{profile}/W_{depletion}$ ratios allows us to estimate our first test criterion from the proportions of the estimates of the depletion method versus PROFILE. The use of ratios is also a way of comparing our results to other studies even though the study design might differ (i.e. sampling depth etc.). -Discussion Line 586-587: The bulk density influences the exposed mineral surface area that is used by PROFILE.

This is one important model parameter (see sensitivity analysis of PROFILE by Jönsson et al., 1995; Hodson et al., 1996). It is unclear how this is an artefact. - Discussion Line 598-610: We think that this section adds relevant information to the interpretation of the results of our manuscript, despite that the inhomogeneity/mixing problem is general for all elements, not only Zr. The mixing erases the depletion gradient that we utilize to quantify element loss. This is an important issue to discuss since the depletion method is widely applied. Whitfield et al. (2011) made an effort to describe Zr variation with depth and highlighted the fact that this is unfortunately rarely done. Contrary to our study, Whitfield et al. (2011) included at least 10 soil profiles where the Zr gradient showed opposite trends, i.e. overall enrichment of Zr in the rooting zone. They related these disturbances to the occurrence of galciofluvial deposits and aeolian till in their study region. In line with Whitfield et al. (2011), we think it is important to highlight which profiles are unsuited for use. In our study we had 7 profiles that fulfilled the assumption of Zr enrichment towards the surface and in one case we used the Ti concentrations since we argued that Zr showed a disturbed gradient. -Line 611-620: We suggest to remove this part, since it is rather speculative. - Figure 3: We agree with the referee and will add a description about how error bars were derived. Minor comments: - By acid silicate bedrock we mean felsic (granitic) bedrock. - Line 94: The referee wrote "recovery is driven mainly by silicate rocks? What about carbonate rocks?" and we answer: We did not detect any calcite in our mineralogical analyses of the soil samples by XRD. Carbonate weathering could be expected in the Caledonian mountain range or from weathering of carbonate-bearing till cose to the caledonian mountain range and confined areas with sedimentary bedrock (e.g. Gotland and Scania). The sites of this studies lie outside the influence of these areas. The bedrock of the Flakaliden site belongs to the Svecokarelian province and the bedrock of the Asa site belongs to the Transscandinavian Granite-porphyry belt ( Fréden, 2009). The study by White et al. (1996) highlighted the importance of small amounts of calcite in intact granitoid rocks and its significance for Ca found in watershed studies. The mean value from the above-mentioned study by White is 0.25 wt. %, and the median 0.075

wt. % calcite. White also noted that in laboratory leaching experiments on the rocks they studied reactive calcite became exhausted after just 1.5 yr. Given the trace concentrations involved and the high solubility of calcite we doubt very much that calcite is or has been of any long-lived significance in the soil profiles studied, even though they are derived largely from rocks of granitic composition. Though we agree that Whites results indicate that calcite in the in-situ granitoid bedrock underlying the soils probably will contribute to Ca export from the catchment. - Line 181-183: We will correct for these spelling mistakes. - Line197-Line 200: Yes, the solid fertilizer mix contained Ca, Mg and K (see table 2 in Linder, 1995). NB! The mass balance estimates of weathering were only based on measurements in the 4 control (no fertilizers) plots. - We agree with the referee and will correct for the following mistakes that the referee has pointed out: "Line 216: Use either 143 C5000 and 10 150 years, or 143 and 10 thousand years. Line 255: space between "and" and "above". Line 267: Figure 5 is cited before Figures 2-4? Can you renumber them? Line 425: Misspelling: Weatherable? Avoid paragraphs with just one sentence, such as those in Line 487-Line 489."

References Scaling and mapping regional calculations of soil chemical weathering rates in Sweden By: Akselsson, C; Holmqvist, J; Alveteg, M; Kurz, D; Sverdrup, H.Water Air Soil Pollut Focus Volume: 4 Pages: 671-681 Published: 2004 Akselsson, C., Olsson, J., Belyazid, S., and Capell, R.: Can increased weathering rates due to future warming compensate for base cation losses following whole-tree harvesting in spruce forests?, Biogeochemistry, 128, 89–105, 2016. Fréden, C.: The National Atlas of Sweden, Geology, Third Ed. SNA Publishing House, Stockholm, 794 Sweden, 2009. Linder, S.: Foliar analysis for detecting and correcting nutrient imbalances in Norway spruce, Ecol. 880 Bull., 178-190, 1995. Olsson M. and Melkerud P.A. (1989) Chemical and mineralogical changes during genesis of a Podzol from till in Southern Sweden. Geoderma 45, 267-287. Simonsson M., Court M., Bergholm J., Lemarchand D. and Hillier S. (2016) Mineralogy and biogeochemistry of potassium in the Skogaby experimental forest, southwest Sweden: Pools, fluxes and K/Rb ratios in soil and biomass. Biogeochem 131, 77-102. White, A. F., Blum, A. E., Schulz, M.

S., Bullen, T. D., Harden, J. W., and Peterson, M. L.: Chemical weathering 696 rates of a soil chronosequence on granitic alluvium .1. Quantification of mineralogical and surface area changes 697 and calculation of primary silicate reaction rates, Geochim. Cosmochim. Acta, 60, 2533-2550, 1996.
* * *

---

## Author Comment (AC2) · 4 Apr 2019

Answer to referee 3: We would like to thank the referee for helpful comments. Different orientation of Fig. 1 and Fig. 5 with depth in the vertical axes: We think that it is ok to keep the figures as they are since the main interest is to show that the concentration/ratios are rather stable. Supplemnetary data: We do not understand what the referee means by "Unfortunately I did not find captions for Supplementary data.", but it might be that he did not have access to the supplementray material. We did however upload it in the submission process, so it should be available. With regard

to the short comments: We will correct for minor flaws in the text. With regard to the referee comment about soil water sampling (i.e. "I am surprised that the soil water was collected only 2-3 times a year"): Normally we have only two sampling occasions per year. The third one is really infrequent.During the 13 years of measurements used in this study soil moisture was sampled once in the spring and once in the fall each year.We could make a supplementary figure to show BC for this period and to compare with the drainage flux.

With regard to all other short comments: We will correct for these minor flaws in the text. Figure 4: We do not think that a removal of site names will improve the figure.

---

## Author Comment (AC3) · 4 Apr 2019

Response to reviewer 1

Reviewer 1 gave very negative opinions of the paper. We have identified three major points that we respond to below.

Lack of any originality / 10 previous papers has shown the same thing It is true that a number of papers have previously compared weathering estimates from the PROFILE model, the depletion method (sometimes named pedon mass balance) and the mass

[Figure]

balance methods (sometimes referred to as the catchment budget method). In this respect our study contributes incremental knowledge to some aspects of this research area. However, most previous studies synthesize estimates from different investigations that, although they are sometimes from the same site, have not been carried out in a harmonized way as regards exact sampling location, input data, method assumptions and scale (pedon/catchment).

The rationale of our study was to perform a detailed harmonized comparative study at two ecosystem experimental sites with particularly good data on three growth and soil nutrients and where the Norway spruce stands were in the phase where growth and nutrient demand is at its highest level. In several other studies, base cation accumulation in biomass and soil change has been assumed to be negligible or zero to facilitate calculations. In situations where base cation uptake in biomass is high, it dominates the flux of the elements included in the balance (here K, Mg, Ca, not Na), and good estimates of the BC uptake are of particular importance. An additional strength of our data is the high quality in the determination of quantitative mineralogy (XRPD data) and stoichiometry (electron microprobe analysis) which is used in the application of the PROFILE model. Furthermore, in our study the mass balance study was performed at the stand/pedon scale with a system boundary in the soil defined at 50 cm depth. This definition of the soil system reflects the root zone and is more relevant for the forest nutrient sustainability estimates than the catchment scale.

We do not therefore concur that similarity of findings with previous studies is an argument for not publishing the study. The inherent difficulties in studying soil weathering means there are still many uncertainties and knowledge gaps. Also findings that corroborates some general features of the methods compared are of great value (see also manuscript by Akselsson et al. in review of this special issue). We look forward to the opportunity to revise the manuscript with a stronger emphasis on a demonstration of the novelty in our study.

Limited sampling We disagree with the critique that the study is based on too limited

sampling data. The Asa and Flakaliden experimental sites are random designs with 4 blocks. The statistical design for this type of experiment dictates that it is the plot mean values that are used in the statistical analyses, not the within-plot variation (it would be pseudo-replication). This means that each plot value is based on a number of subsamples. For example, 25 soil cores were taken from each plot, and soil water was sampled from 3 tension lysimeters per plot. Base cation accumulation in tree biomass was based on site- and stand-specific allometric functions were in total 93 trees at Asa and 180 trees were destructively sampled at different occasions during the period of measurements. In the plots, the incremental growth of each tree was measured repeatedly.

Except for the study by Simonsson et al. (2015) who only estimated weathering rate with the mass balance method at one site in Sweden (and focussed on examination of the uncertainty), we have are not aware of any other similar study on this issue with the same high precision in estimating base cation uptake.

In order to avoid any misunderstandings, we propose to update and clarify the description of the sampling carried out .

Incomplete consideration of uncertainties We agree that this comment is relevant. It was also put forward by reviewer 2. As we have described above, the data available means that the manuscript can be improved in relation to this point.

---

## Author Response (AR1)

[revised manuscript text omitted]

i.e. if current/present-

may differ (Klaminder et al., 2011;

The average  long-term environmental

may differ

the depletion method is

The depletion method makes use of soil profile based mass (Chadwick et al., 1990; Brimhall et al., 1991) to estimate total base cation losses since deglaciation in the soil above a reference soil depth, using an element in a weathering-resistant mineral commonly zirconium (Zr,  present in zircon ) or titanium (Ti,  present in rutile .)

(Sudom and St.Arnaud, 1971; Harden et al., 1987; Chadwick et al., 1990; Bain et al., 1994), due to their stability at low temperatures (Schützel et al., 1963). To yield an annual average weathering rate (mmol, m⁻²), calculated element losses are commonly divided by an estimated number of years lapsed since the onset of weathering.

Steady-state weathering rate may differ from the average during soil formation, which is one reason why weathering rates obtained using PROFILE and the variation in rate with depth in the soil can be expected to differ from those obtained using the depletion method (Stendahl et al., 2013). Observed discrepancies between these methods may therefore reflect 'true' differences or conceptual differences between the methods.

The second approach commonly involves the mechanistic PROFILE model, by which estimates steady state release rates of base cations are estimated using steady-state mass balances based on the dissolution kinetics of a user-defined set of minerals present in the soil, and the physical and chemical conditions that drive the dissolution of minerals. Since it is a mechanistic model, its strength is its transparency, while its main weakness is the difficulty in setting values of model parameters and input variables to which it may have high sensitivity.

Akselsson et al. (*this issue*) concluded that the most important way to reduce uncertainties in modelled weathering rates is to reduce input data uncertainties, e.g. regarding soil texture, although there is also still a need to improve room for improvement in process descriptions of e.g. biological weathering and weathering brakes (e.g. Lampa

Erlandsson Lampa et al., *this issue*). The sensitivity of PROFILE to variations in soil physical parameters (e.g.

soil texture, soil bulk density) and mineral composition wais discussed by Jönsson et al. (1995) and Hodson et al.

(1996), while the importance of the ability to determine the precise identity and quantity of the minerals wais analysed by Casetou-Gustafson et al. (*this issue*).

An The third alternative approach to estimating weathering rate is the base cation budget approach mass balance method (Velbel, 1985; Likens et al., 1998; Velbel, 1985). This methodIt has been applied to estimate current weathering rates at various temporal and spatial scales, and components elements of the budget approach mass balance have been used in different ways in some popular models, e.g. MAGIC (Cosby et al., 1985). The w Using thebase cation budget based on the mass balance mass balance approachW, weathering rate is estimated indirectly as by the base cation budget approach from the the difference between other sinks and sources of elements base cations, which are measured within a system with defined boundaries. The missing source in the mass balance equation is assumed to contain represent the weathering, but can also contain other unidentified sources. The base cation budget mass balance approach is most reliable when based on long-term data from well-defined systems, although even then estimateds of weathering rates tend to suffer from large uncertainties, as errors in the sinks and sourcescan be expected to accumulate in the mass balancee equation (Simonsson et al., 2015). The bIn situation where the base cation uptake in biomass is high, it dominates the flux of the elements included in the base cation budget. NeverthelessFurthermore, the Bbase cation budgets mass balance approach has rarely mostly been applied under non-steady state conditions where accumulation in biomass were not directly measured but estimated to be small, or base cation BC stocks in the soil were assumed to be at steady-state (i.e. including measures of soil exchangeable pools), due to lack of long-term data on base cation fluxes(e.g. Kolka et al., 1996;

Sverdrup et al., 1998; Whitfield et al., 2006). Consequently, at the pedon scale, the PROFILE model and the depletion method are the most frequently used methods in Sweden to estimate weathering rates. The benefit of comparing these two methods is that, taken together, they can provide complementary information about soil
weathering potential (i.e. historical versus steady state weathering) in individual soil layers.

In this study, we applied the our aim was to apply three conceptually different methods of estimating weathering
on , the depletion method, the PROFILE model and the BC budget approach, to two well-defined forest
ecosystems, Asa and Flakaliden in southern and northern Sweden, to allow a harmonized comparison of methods,
and to place weathering in the context of other base cation fluxes in aggrading Norway spruce forests. These
optimum (tree) nutrition experimental sites were selected for the study to obtain a well-defined system with
uniform spatial scale for observations, in line with the suggestion ofby Futter et al. (2012). In addition, BCThe

[revised manuscript text omitted]

(4)equation (4). The confidence intervals for soil change and leaching was based on measured between-plot variation (n=4) using 1.96 as coverage factor. Standard errors The confidence interval for deposition and base cation biomass accumulation in biomass and deposition waswere based on calculations from Simonsson et al.

(2015; Table 4-8 in that paper) of an experimental study site in southern Sweden (Skogaby), since $L_i$ and $\Delta B_i L_{BC}$

and $B_{BC}$ data were not available for individuals plots. A coverage factor of 3 was then used.

**3. Results**

**3.1 Depletion method estimates of historical weathering rates**

At both Asa and Flakaliden, historical weathering rates estimated with the depletion method ($W_{depletion}$) were highest in the upper soil layers and showed a gradual decrease down to the reference depth, which was defined in most plots at 60-70 cm at Flakaliden and for most plots at 80-90 cm at Asa (Fig. 32). Weathering had also taken place below the reference depth. In line with the younger age of the soils at Flakaliden (indicated also by higher abundance of thehad- a higher content of more easily weatehrbale weatherable minerals amphibole, trioctahedral phyllosilicates and calcic plagioclase), and  also athe 
[revised manuscript text omitted]

yr$^{+}$) of weathering rates calculated by the BC budget approach. The combined uncertainty is the sum of 95%
confidence intervals of each term in the BC budget (Eq. 4).

| Site | Element | Deposition | Soil change | Biomass accum. | Leaching | Combined standard uncertainty | $W_{budget}$ | Confidence interval (combined standard uncertainty × 3) |
|---|---|---|---|---|---|---|---|---|
| Asa | Ca | 1.1 | 12.9 | 19.5 | 3.2 | 24 | 58 | ±71 |
| | Mg | 1.1 | 0.6 | 2.5 | 1.6 | 3 | 29 | ±10 |
| | K | 0.3 | 1.0 | 9.7 | 0.1 | 10 | 37 | ±29 |
| | Na | 4.0 | 0.9 | 0.0 | 5.1 | 7 | 7 | ±20 |
| Flakaliden | Ca | 0.8 | 10.5 | 13.3 | 0.7 | 17 | 28 | ±51 |
| | Mg | 0.3 | 1.1 | 1.5 | 0.3 | 2 | 12 | ±6 |
| | K | 0.2 | 0.6 | 6.7 | 0.2 | 7 | 19 | ±20 |
| | Na | 0.7 | 1.2 | 0.0 | 0.8 | 2 | 2 | ±5 |

| Site | Element | Deposition | Soil change | Biomass accum. | Leaching | Combined uncertainty |
|---|---|---|---|---|---|---|
| Asa | Ca | 3.4 | 25.3 | 58.5 | 6.3 | 93.5 |
| | Mg | 3.2 | 1.2 | 7.4 | 3.1 | 14.9 |
| | K | 0.9 | 2.0 | 29.2 | 0.1 | 32.3 |
| | Na | 11.9 | 1.73 | 0.0 | 10.0 | 23.6 |
| Flakaliden | Ca | 2.5 | 20.6 | 40.0 | 1.4 | 64.4 |
| | Mg | 0.9 | 2.2 | 4.6 | 0.6 | 8.3 |
| | K | 0.5 | 1.2 | 20.2 | 0.4 | 22.2 |
| | Na | 2.2 | 2.4 | 0.0 | 1.6 | 6.1 |

**Figure captions**

**Figure 1.** Titanium (Ti) to zirconium (Zr) ratio (by concentration) used as an indicator of uniform parent material in all soil layers at Asa (F3, F4, K1, K4) and Flakaliden (10B, 11B, 14B, 15A).

**Figure 2**. Zirconium (Zr) gradient in the soil at Asa (KI, K4, F3, F4) and Flakaliden (10B, 11B, 14B, 15A)

**Figure 3**. (Left) Historical weathering rate of base cations (mmol$_c$ m$^{-2}$ yr$^{-1}$) estimated by the depletion method and (right) steady-state weathering rate estimated by the PROFILE model in different soil layers at Asa and Flakaliden.

**Figure 4**. Comparison of weathering rates (mmol$_c$ m$^{-2}$ yr$^{-1}$) for Ca, Mg, Na and K determined with the depletion method, the PROFILE model and the base cation budget method for the 0-50 cm layer at Asa and Flakaliden. For the weathering rates based on the depletion method and the PROFILE model, error bars represent the standard error calculated based on four soil profiles at each study site, except for Flakaliden, where the depletion method was only applied in two soil profiles. For weathering rates based on the base cation budget approach, error bars represent combined standard uncertainties, which are based on standard errors derived from plot-wise replicated data of the present experiments (for leaching and changes in exchangeable soil pools) and on standard uncertainties derived from Simonsson et al. 2015, where replicated data were missing in the present study (for accumulation in biomass and total deposition).

**Figure 5.** (Left) Sources and (right) sinks of base cations in ecosystem net fluxes at Asa and Flakaliden. The soil is a net source of soil base cation stocks decrease and a net sink if they increase. 'BC budget' = current base cation weathering rate (W$_{budget}$) estimated with the base cation budget method, including measured changes in soil extractable base cation stocks; 'PROFILE' = soil extractable pools estimated from base cation budget using PROFILE estimates of steady-state weathering rate; 'Historical' = soil extractable pools estimated from base cation budget using estimates of historical weathering rate by the depletion method. 'Measured soil change' and 'Base cation budget estimated soil change' indicates that equation 4 was used to estimate weathering rate or the soil change, respectively.

**Figure 6.** Time (years) required to achieve the measured historical element loss in different soil layers with maximum or minimum PROFILE weathering rates at (a) Flakaliden and (b) Asa.

[Figure]

[Figure]

**Figure 1.**

[Figure]

**Figure 2.**

[Figure]

**Figure 3̶2.**

[Figure]

[Figure]

[Figure]

**Figure 4̶3.**

[Figure]

**Figure 54.**

[Figure]

[Figure]

[Figure]

[Figure]

**Figure 6.**

Dear Prof. Andersson,

Thank you for providing the reviews and for your editorial comments on our manuscript and the opportunity to submit a revised version. Your suggestions (marked in grey color) were:

Original comments:

Associate Editor Decision: Publish subject to minor revisions (review by editor) (12 Apr 2019) changed to major revision (review by editor) (15 Apr 2019) by Suzanne Anderson Comments to the Author:
I have received three reviews of the manuscript, of which two were quite positive and one was extremely negative. My own reading lies somewhere between these poles. I think with attention to the comments of the reviewers the authors will be able to produce an acceptable contribution.

The goal of the manuscript is to assess different methods to determine current mineral weathering rates in soils. The question is of more than academic importance, as mineral weathering must in the long run replace cation loss resulting from forest harvesting and consequent soil acidification. The work is conducted in the context of a Swedish environmental goal of managing sustainable forestry. Three established methods of assessing weathering rates are tested in two well-characterized forest soils in Sweden. While the work is clearly important from this perspective, the question and the methods used are not an approach that is likely to bring about a paradigm shift or blinding insight.

Reviewer #1 offered the very negative appraisal, based mostly on two points—first that the methods and models used have been compared in other sites before, and therefore there is nothing new. The second complaint was that little data was used from the two study sites. I sympathize with the authors response that this study is distinguished from others by the quality of the data available for the method comparison (a key being uniformity of collection and analysis methods). Both negative points can be addressed with nearly the same antidote: providing a much clearer description of how the data used is generated. I suggest the authors come up with a way to highlight responses to this reviewer very early and with some panache in the text. The text is rather long overall, and while methodical in its presentation, it is easy to miss the point amidst some of the details.

General points to attend from Reviewers 2 & 3:
1) Terminology: Reviewer #2 found the manuscript very interesting, but also offers some helpful guidance on improving clarity. I concur with this reviewer that the terminology naming the methods was hard to keep straight. In short, these are "depletion method", the "steady-state method", and the "mass balance method". For instance, the depletion method (based on the assumption of Zr immobility) is in essence a mass balance calculation. I wonder if the authors could come up with a small table that would summarize the 3 methods? It would be helpful to have these briefly outlined and set apart from the text in a way that the reader can circle back quickly when confusion sets in.

I believe some terminological confusion arises because the goals of this study are narrowly focused on determining the weathering rate going on in Swedish forests now, to address the needs of forest management now. Some of the methods employed (depletion method in particular) are averaged over much longer timescale than the present (more on this below). Moreover, choices such as focusing only on the top 50 cm are driven by interest in weathering in the rooting zone. I think the authors might rewrite the introduction and parts of the discussion to highlight how their motivations might differ from those of a pedologist, interested in the full history and depth of soil development, or from a geochemist interested in weathering rates over long time scales.

2) Soil age: I disagree with Reviewer #2 on soil age changing with depth in the soil profile. It is standard soil science to use soil age in the manner used by the authors. However, the authors need to correct an error in their reporting of soil ages. Asa is reported to be 143 000 thousand years old (or 1.43 x 10^8 years…. Cretaceous!), while Flakaliden is reported as 10 150 thousand (1.015 x 10^7 years). I'm pretty sure the authors mean 143 ka and 10.15 ka, respectively. These ages are significantly different. Asa has been exposed and weathering since the last interglacial, while Flakaliden is only Holocene. The significant difference in weathering (disintegration of boulders, accumulations of clay and organic matter, etc) in till from different glaciations is an important tool in determining relative ages. I am surprised that the influence of the much longer pedogenesis at the Asa site is not addressed in the manuscript. Since Asa was apparently not glaciated in the last glaciation, I would guess it was instead subjected to periglacial processes for ~100 ka of its age. I also wonder about any eolian (e.g. loess) deposition during the last glaciation? In any case, I would like the authors to address the very different history of weathering at these sites and how this impacts the weathering rates they assess. (In line 218-219, the soils are described as similar in B horizon thickness-- quite surprising given the 1000 fold difference in age of these soils that the B horizons are similar in thickness. You will need to explain why-- it's striking that the very old soil in the warmer climate seems to be quite similar to the very young soil in the colder climate.)

3) Table 4 "quality of term quantification". Several reviewers were unimpressed with a vague and subjective description of the quality of quantification of different terms in the mass balance method. Is there a better way to do this?

Reviewer #2 offers a number of useful comments on minor points in the manuscript, please address these.

4) I found the Conclusions section to be unsatisfying. As the 11th group to compare different weathering rate computation methods, is our knowledge of weathering rates improved? Did using more carefully collected data and uniform sampling methods improve these numbers? I am not sure from reading the conclusions. Nor am I sure what recommendation would be made to land managers on the basis of this work for how to measure weathering rates. I suggest pulling back out to the big picture, and addressing the reader who has skipped from introduction straight to conclusions in rewriting this section.

My own minor comments:
Line 65: The definition of "weathering rate" given does not define a rate (dimensionally).
Line 182: I'm surprised at lack of attention to till mineralogy: does this differ between sites?
Line 200: Describe the solid fertilizer used.
Line 218: "sandy loam till" is not a texture. "sandy loam" is.

Line 607-608: Pinus is conifer, not deciduous. What was the vegetation at Asa during the last glaciation?

Response:

We have carefully read through all the comments and have critically discussed possible changes. We hope that you will find all major points sufficiently addressed and that the manuscript is acceptable to you in revised form.

In terms of changes we have focused mainly on your principal suggestions, i.e. to "describe very early and with panache" that three methods are required to cover three very different conceptual views on weathering (i.e. steady-state, historical and dynamic weathering). To our knowledge this has not been done to the same extent previously based on harmonized implementation of all methods for the same sites. Kolka et al. (1996) is the only group that used a similar approach, however, that study was based on less detailed data. We hope that our discussion of these different estimates in a mass balance perspective adds further novel insights into how these methods relate to other ecosystem sources and sinks of base cations in an aggrading forest ecosystem.

Details of the changes in the manuscript are as follows :

1. Terminology:

- Throughout the manuscript: We have changed the name of the third method used to estimate weathering from "mass balance approach" to "base cation budget". We have also added a new table (Table 2) that summarizes major differences (conceptual, scales etc.) between the methods.

- Additionally, we have rewritten parts of the introduction and discussion to highlight our rationale behind the use of the perspective of historical versus present-day weathering estimates.

2. Soil age (Line 284): The soil age at Asa should be 14,300 years, not 143,000 years. This was an unfortunate typo in the text (but the correct age was given in Figure 6). The soil age at Flakliden is younger, but both soils were formed during the end of the latest (Wechselian) ice age, and there is no eolian loess deposition at either site (Fréden 2009).

3. Former Table 4 (Now table 5:  We have now attempted to give an additional description of the quality and uncertainty of different terms in the base cation budget (former "mass balance method") by estimating combined uncertainty, which is given in a separate Table (Table 6). The combined uncertainty provides a better picture of the uncertainty of weathering estimates by the base cation budget approach.  New text describing the method of calculation, material and methods, results and discussion is now inserted at appropriate places in the manuscript, i.e., at Lines 582-595, 700-706  and 945-946.

4. Conclusions: We have made a new improved version of the conclusions that we hope is clear and informative with respect to the novelty of our study, and has more obvious links with the context and rational given in the Introduction. We stress that a contribution of new knowledge is gained from using a harmonized study design, i.e. (1) where similarities and dissimilarities of methods occur with regard to different test criteria, and (2) in terms of the similarities and dissimilarities of different methods with regard to their relative importance in the overall base cation budgets.  All of which enable us to identify some important issues for future research.

5. Minor comments:

-Line 66: We have changed the definition of weathering rate so that dimensions are evident.

-Line 247-249: We have cited Casetou-Gustafson (2018) where all details can be found about soil mineralogy at Asa and Flakaliden.

-Line 267-268: We have added the information about which base cations were contained in the solid fertilizer mix.

-Line 287: We changed soil texture to sandy loam.

-Lines 899-900: We improved this sentence. Furthermore, we would like to clarify that there was no vegetation during the last glaciation as Asa was glaciated.

Additional changes:

- We found an unfortunate error in the calculation of fine-root biomass which is now corrected (Line 505). The change has no effect on how the calculation is described in the Materials and Methods section, but the consequences is a reduced fine root biomass leading to ca 5% lower BC uptake in biomass for Ca, Mg and K. This had no impact on the conclusions and the general picture, but the substantial discrepancies between present-day weathering rates produced by the base cation budget and PROFILE are now slightly less pronounced. The new calculations are now included in the revised
Figures 4 and 5 (former Fig. 3 and 4).

- The description of how base cation budgets were calculated is now expanded with detailed
information about sampling procedures and sampling sizes (section 2.5), as was requested by several
reviewers. Additional graphs showing temporal changes in soil water chemistry and runoff are
included in the Supplementary material (Figure S4-S5; see also below). Apparently, reviewer 1 in
particular got the impression that the base cation budgets were based on a quite limited sampling.
We hope the present text will change that view.

-By mistake error bars in Figure 3 (now Fig 4) have been switched between K and Na and we have
corrected for this.

-We have revisited our results that we obtained from applying the depletion method and came to the
conclusion that the relatively low values given by the historical weathering losses is due to
complicated Zr gradients observed in two soil profiles at the Flakaliden site (i.e. 11B and 15A; Fig.5).
By removing these profiles from the calculations (described now in Lines 363-367 and Line 904-917),
predictions of historical weathering losses for Ca and Mg are more in line with the general view of
declining rates over time. It makes more sense to observe Ca and Mg losses that are of similar
magnitude, since both elements are contained to a similar extent in the easily weatherable mineral
hornblende, which is an important mineral in the Svecofennian granitic bedrock. An editorial
consequence of this was that we also moved former Fig. 5 to Fig. 2.

**Referee 1:**

-In order to strengthen our study we have attempted to be clear about the novelty from the beginning
of the manuscript (i.e. Introduction). Furthermore, as described above, we have added an additional
assessment of uncertainty of the data that are used to construct the base cation budget (Table 6).

**Referee 2**:

Historical weathering:

- We have corrected for the wrong unit used for Zr in former Table 2 (now Table 3).
- We have added an equation (1) to the manuscript in the method section which describes the
calculation by the depletion method (Line 327).
- We have added estimated volume change (strain calculations according to a formula in White
et al.1996) in the Supplements (Table S5) and a mentioning in Lines 348-352.
Steady-state weathering:

- We included a short mentioning of the steady-state concept in the introduction (Line 148) and in Table 2.
- We have enlarged former Table 3 (now Table 4) and added a column that contains general information for each site. Layer-specific information is given in Supplementary Table S3-S4.

Base cation budget:

- We have added a more detailed description of how total deposition was calculated. The same deposition data was used for parameter setting of PROFILE and in the base cation budget.
- We have added two supplementary figures (Figure S4-S5) to illustrate monthly mean and standard deviation of drainage (mm) in soil water at 50 cm depth in the soil of the four control plots at Asa and Flakaliden (Figure S5) and mean and standard error of BC ions (mg/l) in soil water sampled at 50 cm depth in the soil of four control plots at Asa and Flakaliden (Figure S4).
- We have added a new table to the manuscript (Table 6) that contains estimated uncertainty ranges for the different terms in the base cation budget and their combined uncertainty. These uncertainties are based on standard errors derived from plot-wise replicated data of the present experiments (for leaching and changes in exchangeable soil pools) and on standard uncertainties derived from Simonsson et al. 2015, where replicated data were missing in the present study (for accumulation in biomass and total deposition). A detailed description of the statistical procedure is given in Lines 582-595.

Discussion:

We have improved section 4.2.

**Figures:**

- We have inserted an explanation of error bars in Line 1494-1500 and in Lines 582-595.

**Referee 3**

- We have removed site names in figure 4 (now figure 5).
- Apart from issues that were already raised by the other referees, we have corrected for minor errors, such as misspellings and we have changed to "extractable concentrations" instead of "concentrations" in the table description of former Table 2 (Now Table 3).

In addition to the revised manuscript we have also provided a corresponding version with tracked changes on, so that the revisions are clearly visible. We have also been carefully through the whole text and made some additional minor revisions which we believe aid clarity.

We hope you will find our revised manuscript acceptable for publication in Biogeosciences and look forward to hearing from you in due course.

On behalf of all authors, yours sincerely,

Sophie Casetou-Gustafson,

Corresponding author

---

## Referee Report (RR1)

**Review of** *Current, steady-state and historical weathering rates of base cations at two forest sites in northern and southern Sweden: A comparison of three methods* **(v3, 1ˢᵗ revision?) by Casetou-Gustafson et al. BG-2019-47 by Michael Velbel**

*The science*

1.  This is a paper about weathering rates in forested landscapes on glacial parent materials that only a specialist can love. The slog through under-and mis-referenced statements, mis-numbered Tables, and almost obsessively detailed text was a journey in which not every reader will persevere, but it forced this reviewer to more or less see the whole picture – albeit not without difficulty. The paper will likely be more widely read by novices and other non-specialists who will be drawn to the shiny inter-approach comparison without the benefit of extensive experience interpreting either outputs of individual approaches or comparison of such outputs from multiple approaches. Such readers will either cite the paper for the nuggets they understand from the densely written text or be flummoxed by all the highlighted details and nuances. But that's their problem.

2.  Strength (mostly) with some weakness: Some descriptions of pioneering previous work are more correct, and therefore more useful to readers, than others. The "base cation budget" approach as developed in the two papers cited in lines 157-158 is a steady-state linear inverse model. So far so good. However, no aspect of MAGIC is an inverse model (base-cation influx-outflux ("input-output" model). MAGIC – as thoroughly described in the OTHER Cosby et al. 1985 Water Resources Research paper, but not in the Cosby et al. 1985 WRR paper cited in this manuscript – is a forward model in which the (bulk, catchment-averaged) "weathering" rates of the four major base cations are input parameters (see input definitions in the last section of their Table 1).

    Later in the manuscript, lines 559-566 are is well-referenced to several classic references correctly represented, and in that sense invokes insights from (although contributes none to) inter-regional/global literature on weathering rates in forested landscapes on glacial parent materials in general.

    Overall, some misrepresentations and omissions of classic works notwithstanding, this study takes good advantage of pioneering insights of general (global) significance for interpretation of the local weathering rates estimated by diverse means.

3.  Weakness: Lines 145-155 - Sensitivity of PROFILE outputs to slight variations in input parameters is correctly acknowledged. What is not discussed is the sensitivity of PROFILE outputs to the mineral dissolution rates and rate laws used in the forward modeling.

4.  Strength: Lines 269-286 – The number of profiles rejected for not meeting reasonable acceptance criteria on the one hand suggests reasonable and useful criteria. On the other hand, failure to satisfy many of the numerous assumptions acknowledged disqualifies a distressingly large number of profiles. This suggests that an approach with so many assumptions that so many natural systems (in this case, profiles) cannot satisfy may not

be an approach that reliably produces outputs suitable for comparison with other approaches. Such comparisons being the whole point of this manuscript, this paragraph undermines reader confidence in the entire enterprise. Fortunately, later in the manuscript, precisely these limitations of the assumptions are explicitly addressed and considered in interpreting the core truths hidden amidst the various model outputs.

5.      Lines 287-288 – Was bulk density *measured* for each soil layer except in some plots where density measurements could not be made below a certain soil depth? Line 287 says estimated. If estimated (as written), then how was the estimation in line 287 different from the estimation in line 288ff?

6.      Strength: Section 2.5 – all the components of the base-cation budget seem to be well-constrained. Much good discussion later in the manuscript benefits from this thought and care.

7.      Weakness: Lines 438-443 – Using outputs of the other two approaches to constrain uncertainty in the budget approach breaks down the wall of independence, and therefore the validity of comparisons, between the three approaches.

8.      Weakness: The treatment of error is confusing. Eight plots were sampled, four in each of two study areas. At each study are, two control plots and two fertilized plots were sampled. Both fertilized plots at Flakaliden were "eliminated from further consideration in calculations of historical weathering rates using the depletion method" (lines 282 - 283). Site mean or average values and their standard errors or combined standard uncertainty were calculated for each of the three approaches. "For the weathering rates based on the depletion method and the PROFILE model, error bars represent the standard error calculated based on four soil profiles at each study site, except for Flakaliden, where the depletion method was only applied in two soil profiles." (Lines 1045-1048, caption for Figure 4). However, error bars are shown for all cations for the depletion approach applied to Flakaliden; the reader is left to infer that these error bars show the range (not the SEs) of the rates estimated for the (only) two Flakaliden profiles.

9.      Strength (good inferences) and weakness (text is under-referenced w.r.t. relevant past work elsewhere): Lines 500-501 – "The closest resemblance between methods was found between $W_{depletion}$ and $W_{budget}$ for Na." No surprise here; Taylor and Velbel (1991) showed that diverse approaches (in their case, excluding versus including biomass) yielded more similar weathering rates for Na-hosting minerals than for the mineral hosts of other base cations, because (as is also noted in this manuscript in lines 504-505 and 657-658), the independent variables (input terms to the inverse model) for Na budgets are the least influenced by variations and uncertainty for biomass stocks and flows relative to the corresponding terms for any other base cations. This enables intersite comparisons of weathering rates for Na-host minerals more readily than for the mineral hosts of any other base cations (Price and Velbel, 2005). Similarly, Taylor and Velbel (1991) and Velbel (1995) showed that estimated weathering rates of K-hosting minerals are the most influenced (relative to the hosts of other base cations) by the large role botanical demand for K and the corresponding large sensitivity of calculated rates to quantification of

biomass-related concentrations and fluxes. Lines 504-506, 657-658, and 664-666 of the present manuscript rediscover this phenomenon effectively and concisely. This correspondence between the work presented in this manuscript and (unfortunately uncited) long-known previous work inspires reviewer confidence in this manuscript's results from using the cation-budget approach.

10. Strength (good inferences) and weakness (text is under-referenced w.r.t. relevant past work elsewhere): The case that solute budgeting is the most reliable approach has been previously made (Velbel and Price, 2007). The sound results of the cation budget approach in the present manuscript, and the weak correspondence between the results of the other two approaches with the cation budget results (Fig. 5 in the present manuscript), both appear consistent with this previously articulated argument insofar as consideration of weathering rates is concerned.

11. Strength: In light of the previous several comments, I find Discussion section 4.1 Comparison of conceptually different methods (lines 522-540) an honest statement of how all approaches have their individual limitations, and a sound (although I cannot vouch for pioneering or up-to-date) assessment of the state-of-the-science regarding how to use the different approaches for scientific investigation. The rest of the Discussion expands successfully on section 4.1 in considerable detail. (The implications of all this for using such approaches for decision support in land-use management is another matter entirely.)

12. The effort to detect a pulse in PROFILE is both charming and valiant.

13. Weakness (text is under-referenced w.r.t. relevant past work elsewhere): The depletion method rests weightily on the assumption of Zr immobility. The assumption of Zr immobility is well-justified on crystal-chemical grounds, and consistent with observations and consequent inferences in many studies. However, unfortunately, Zr immobility may be an actual extant limiting condition in fewer weathering profiles than the number to which the assumption has been applied. In addition to the parent-material (glacial till) heterogeneity and redistributive processes in plow-layers and by earthworms that are invoked in this manuscript (lines 607-633), there exist papers (albeit only a handful) over the past four decades or so demonstrating both corrosion of zircon in soil grains and physical translocation of fine-grain zircon through soil-regolith pore networks. Unfortunately, I have not yet worked systematically on Zr as a mobile versus reference element and have thus not had to recover this literature, and in some of the papers the relevant observations are incidental to the main point of the paper, so I would be able to put my hands on only one of these papers at present.

14. Strength: The comparisons of estimated total depletion that would follow from the weathering rates determined by the various approaches with the measured total depletion over the duration of pedogenesis since deglaciation (Figure 6) is novel, and is only possible because so much work has been done at the study sites in support of the work reported in this manuscript and the related papers in the same volume.

15. Strength (good inferences) and weakness (text is under-referenced w.r.t. relevant past work elsewhere): Attributing the differences between the three approaches to changes in the exchangeable cation pool is well-considered, and well supported by this study's multiple measurements of cation exchange pools one to two decades apart (lines 361-364). The suggestions for not-yet-measured sources of Ca and K are well-considered, at least to the extent that they conform with this reviewer's experience with how complicated the mass budgets for these specific elements can be (e.g., Taylor and Velbel, 1991; Velbel, 1995; Price et al., 2005).

16. The data, observations, evidence, and inferences presented in this manuscript support a for the most part persuasive case for the stated interpretations, conclusions, and modest implications for similar work in similar landscapes.

17. If I ever get to teach my graduate-level "weathering" course again (increasingly unlikely as my career careens toward an untimely collision with retirement), I would assign this paper as either an introduction to the three approaches or, after the three approaches had been individually introduced from older pioneering studies, as the starting point for a treatment of their inter-comparability, relative scholarly merits, and relative utility.

*The manuscript*

18. The work as presented appears to be sound but the presentation of the work needs to be improved by better explanation of observations, relations between observations, how the observations support the inferences and interpretations, *and their implications for the literature*. Revisions of the manuscript are required before further review and an editorial decision.

Regarding the organization of the scientific content in the manuscript: Specific matters relating to this manuscript are highlighted in the following numbered items and the subsequent line-index list.

19. Clear identification of scientific novelty in Introduction (Statement of Purpose): A successful manuscript must identify the novelty of the work reported in terms of advancing the frontiers of forest-soil weathering. A successful manuscript should clearly identify what is new and scientifically significant about the work presented. The authors should clearly state the aim of this study (i.e., what problem you want to solve, etc.). This will help readers to understand the scientific novelty of this study. The authors need to show (state clearly) how this study differs from past studies of similar soils. The Introduction of the paper should define the scope of the problem to be solved by the work presented. The Introduction should provide background explaining what problem is being solved, or what gap in existing scientific-community understanding as reported in the literature is being filled, by the work reported in the paper. The Introduction should briefly explain why this problem is not already solved or why the solution in this manuscript is better than previous solutions in some specified important way.

The Introduction section makes a number of good and promising points:

Lines 114-117 - "Differences in input data can be attributed to different time scales used when acquiring different input data, challenges determining accurate mineralogical compositions and the use of field data compared with laboratory data (Van der Salm, 2001; Futter et al., 2012). Thus, they recommend that at least three different approaches be applied per study site to evaluate the precision in weathering estimates."

Lines 120-121 - "The choice of methods is primarily based on the fact that rates of weathering may vary over time (Klaminder et al., 2011; Stendahl et al., 2013)."

Lines 140-142 - "Since the rate of weathering may vary over time (Klaminder et al., 2011; Stendahl et al., 2013), the average 'historical weathering' rate may differ from the present-day weathering rate. The depletion method is most widely used in Sweden to estimate weathering rates, specifically at the regional scale (Olsson et al., 1993)."

Lines 162-164 – "The base cation budget approach is most reliable when based on long-term data from well-defined systems, although even then estimated weathering rates suffer from large uncertainties, as errors in the sinks and sources accumulate in the mass balance equation (Simonsson et al., 2015)." The case has already been made elsewhere that solute budgeting is the most reliable approach (Velbel and Price, 2007).

Lines 165-167 – The manuscript is correct in observing that "The base cation budget approach has mostly been applied under conditions where accumulation in biomass were not directly measured but estimated to be small, or base cation stocks in the soil were assumed to be at steady-state (e.g. Kolka et al., 1996; Sverdrup et al., 1998; Whitfield et al., 2006)."

Lines 173-178 – "The base cation budgets were estimated at the period of stand development when nutrient demand was expected to peak. In combination with access to highly accurate data on biomass production, these conditions also provided opportunities to relate weathering to base cation accumulation in biomass at high nutrient uptake rates, and possible simultaneous depletions of extractable base cation stocks in the soil. Furthermore, input data to PROFILE were characterised by high quality quantitative mineralogical data, measured directly by X-ray powder diffraction (XRPD), as previously discussed by Casetou-Gustafson et al. (2018)."

So far, so good.

20. If the purpose is primarily to describe a local phenomenon, then either the paper belongs in a local or regional journal or the manuscript needs to better explain the scientific novelty of the work. Does this study of these phenomena and models at these places have significant potential to change our understanding about such phenomena and the application of these models elsewhere, or does it only confirm that it is much like others that have been similarly characterized and modeled and it is just adding information about this particular occurrence? How does the detailed study of these two field sites

advance scientific understanding beyond merely characterizing this locally important land-use type? If there is no larger scientific significance, then specifically how does the work reported improve the management and use of the forest resource? A study of sites and with methods that do not have international or global significance for understanding that type of deposit is not self-evidently sufficient for an international journal. Is the work reported an ensemble of measurements and tasks that has significant potential to change scientific or applied approaches to this kind of landscape? What is distinctive or unique about the reported work relative other published approaches to the same problem? What can the community do better after the work reported here than it could do without this work? Specifically how does the work reported in this manuscript improve upon prior art? I look forward to further explanation about this, and especially for this to be very clearly stated in the Introduction and Abstract of the revised version.

21.   The Introduction should set the stage for the Discussion and Conclusions sections: Once the Introduction and statement of purpose have been written (as discussed above), the Discussion and, especially, the Conclusions, should clearly and explicitly link the outcomes of the research with the identified gap in community understanding as identified in the Introduction. The present manuscript as written does not accomplish this. The mere fact that specific sites have not been previously examined for the reported phenomenon, or a specific combination of characterization methods and modeling approaches has not been previously applied, does not by itself justify publication in an international journal, and Conclusions that do not link the Results and Discussion back to the literature on a larger scientific problem are not enough to justify publication in an international journal.

22.   A description of a new soil-weathering case study must thoroughly compare the data from the new deposit with data from the literature (primary papers are best, but review papers may suffice) on similar landscape/land-use types elsewhere. This allows the reader to appreciate the paper's inferences about what is new or unusual about the study area, and what is familiar and common about it. The present manuscript as written does not accomplish this, which may be OK considering that it appears not to be intended to. The first comparative assessment (lines 548-557) and much of what follows in section 4 is local-regional comparison with other Swedish studies back at least as far as 1995. This content comes across to a reader unfamiliar with recent studies in Sweden as detailed and thorough, albeit perhaps not especially exciting. The second comparative assessment (lines 559-566) is well-referenced to several classic papers, and in that sense invokes insights from (although contributes none to) inter-regional/global literature on weathering rates in forested landscapes on glacial parent materials in general. This study takes good advantage of such pioneering global insights for interpretation of the local weathering rates estimated by diverse means. However, the work reported appears to be of local-regional significance only. It provides no major new insights for forested landscapes on glacial parent materials in general. If the purpose of the paper is primarily to describe a local deposit without such a comparison with other similar deposits, then the paper may be more appropriate for a local or regional journal, economic geology journal, or trade journal that publishes local resource-evaluation work, rather than an international journal.

23. A locality map is a near-universal courtesy to readers. This manuscript requires one. The fact that it lacks one sadly reinforces the perception created by the rest of the manuscript, that it is directed mainly to readers who already know where the study sites are and why they should care.

24. Why is Fig. S3 the first Figure cited (even before S1 and S2)? All Figures should be numbered in the order they are first referred to in the text, and referred to in the text in their numbered order. The manuscript as written is poorly organized in this regard. Please make the necessary adjustments so that all Figures are cited in their numbered order, and numbered in their cited order.

25. All Tables should be numbered in the order they are first referred to in the text, and referred to in the text in their numbered order. The manuscript as written is poorly organized in this regard. For example, Table 1 is cited in line 204, but the information described in the corresponding text is on Table 4a. Please make the necessary adjustments so that all Tables are cited in their numbered order, and numbered in their cited order.

26. This manuscript is seriously under-referenced. Numerous statements are made in such a way that the reader cannot distinguish statements about common knowledge among Swedish forest-soil scholars from statements that ought to refer the reader back to an authoritative primary reference. Also, pioneering primary references, the original sources of ideas, should be cited, rather than citing more recent derivative papers that apply the same concept or say the same thing (e.g., regarding use of Zr vs. Ti as a reference element).

27. Supplemental information: To the extent that the large tables (e.g., Tables S1 and S2) are essential to the paper, the Editor and authors might consider making the same tables available in Excel or .csv form as supplemental online material. Data available in a directly usable format (not requiring transcription or OCR from the printed/pdf version) might wind up being widely used by the community.

Regarding the English, I commend the authors for a rather well written manuscript. I did note, however, that the English style and format does not quite adhere strictly to expectations. I have listed below some specific points to which attention should be directed (e.g., typographical and grammatical errors, word choices, punctuation, capitalization, sentence structure, subject-verb agreement, paragraph organization, &c.).

28. Generally speaking, use "since" only when referring to time rather than as a conjunction in place of "because." Several occurrences (line 140, 2$^{nd}$ occurrence; 147, 387, 516, 536. 560, 630, 666, 668, 673, and 690) require correction in this regard.

29. Use "by" or "by way of" or "by means of" rather than "via" ("via" connotes a spatial, geographic pathway or route, and is not to be used metaphorically) or "through" (which Lalso has spatial connotation). Line 599 requires correction in this regard.

30.     Replace "frequent(ly)" with "commonl(ly)" in lines 107 and 168.

31.     "Data" is plural; "datum" is singular. Data are/were, not data is/was (lines 401-2, 433); many other occurrences are correct.

32.     The concept of "stability" is not used with sufficient rigor in this manuscript. Any phase (or assemblage) or state is either stable (at equilibrium with one or more specified other phases) or unstable *with respect to a specific alternate phase assemblage or state under a given set of conditions*. This specific state must be explicitly stated (e.g., stable with respect to dissolution in dilute acidic solutions at Earth-surface conditions). A mineral that survives for a long time because it reacts slowly is persistent, not stable (Velbel, 1999). Line 137 requires attention, consideration, and rewriting in this regard.

33.     Replace "stable" with "constant" or "uniform" in lines 274, 631, and 633.

34.     Strata (or beds) are produced by sedimentological superposition of layers of physically mobilized grains deposited from a fluid medium under the influence of gravity. Horizons are produced by the pedogenic equivalent of chromatography. If the origin of discrete depth intervals of great lateral extent is uncertain or indeterminate, neither term should be used. All references to "layers" should be changed to "horizons" or "intervals".

35.     Please replace "x"s used as multiplication signs with the multiplication sign ("×") in lines 194 and 375.

36.     Be careful to match the number (singular vs. plural) of articles, subjects, and verbs. Several occurrences (e.g., line 707) require correction in this regard.

37.     This reviewer did not check the references for accuracy or style, or for conformity between references cited in the text and those listed in the bibliography.

General comments:

There are many places in the text where, alone or in concert, insertion or redeployment of commas and semi-colons may make long sentences easier to read.

Specific comments:

Line 101 – replace "if" with "whether"

Line 170 – replace "on" with "to"

Line 353 – replace "ar" with "is"

Line 416 – Delete close paren.

Line 481 – Replace "as opposed" with "in contrast".

Line 580 – Replace "were possible to reconcile" with "could be reconciled"

Line 648 – delete the duplicate period.

Line 832 – "Sedimentologists" is plural.

Line 864 – Journal title should be in title case (all major words capitalized).

Figure axis labels should be in the format "Label (units)". The experienced reader presumes that elemental "concentration (%)" in Figure 2 means weight %, but, because it could be atomic or molar %, ("wt. %") would eliminate the possibility of misunderstanding by non-specialists and novice readers. The labeling of axes for all other Figures is excellent.

Tables S1 and S2 are not useful as formatted. Graphical representation of the sensitivity analysis is required if it is intended to be understood by readers.

Tables S3 and S1b contain similar data for the two field areas; the numbering of these tables does not make sense.

Table S4 – Reporting model-input soil bulk densities and exposed mineral surface areas to 15 significant figures is not justified by anything explicitly stated in the text.

These comments, above and below, are intended to help improve the effective presentation of the work done and the scientific impact of the revised manuscript.

References

Cosby, B. J., Wright, R. F., Hornberger, G. M., and Galloway, J. N., 1985. Modeling the effects of acid deposition: Estimation of long-term water quality responses in a small forested catchment.  Water Resources Research, v. 21, p. 1591-1601.

Price, J.R., Velbel, M.A., and Patino, L.C., 2005. Rates and timescales of clay-mineral formation by weathering in saprolitic regoliths of southern Appalachian Mountains from geochemical mass balance. Geological Society of America Bulletin, v. 117, no. 5, p. 783-794.

Taylor, A.B. and Velbel, M.A., 1991. Geochemical mass balance and weathering rates in forested watersheds of the southern Blue Ridge. II. Effects of botanical uptake terms. In: Pavich, M.J. (editor), Weathering and Soils. Geoderma, v. 51, p. 29-50. (In the same issue as the Brimhall paper)

Velbel, M.A., 1995. Interaction of ecosystem processes and weathering. In Solute Modelling in Catchment Systems (Trudgill, S., editor), John Wiley & Sons, pp. 193-209.

Velbel, M.A., 1999. Bond strength and the relative weathering rates of simple orthosilicates. American Journal of Science, v. 299, p. 679-696.

Velbel, M.A., and Price, J.R., 2007. Solute geochemical mass-balances and mineral weathering rates in small watersheds: Methodology, recent advances, and future directions. Applied Geochemistry, v. 22, p. 1682-1700.

If other even older papers made the same points as those listed above, the bibliographies of the listed papers may help locate the true pioneer papers.

A Table I produced for my own use in this review.

| Method | Asa site base-cation weathering rates $mmol_c$ $m^{-2}$ $yr^{-1}$ | | | | Flakaliden site base-cation weathering rates $mmol_c$ $m^{-2}$ $yr^{-1}$ | | | |
|---|---|---|---|---|---|---|---|---|
| | Ca | Mg | K | Na | Ca | Mg | K | Na |
| Historical depletion | 4.7 | 3.1 | 0.8 | 2.0 | 11.0 | 12.9 | 3.2 | 7.0 |
| Steady-state forward model | 8.9 | 3.8 | 5.9 | 18.5 | 11.9 | 6.7 | 6.6 | 17.5 |
| Base-cation budget | 65 | 23 | 40 | 6.6 | 35 | 14 | 22 | 2.2 |
| Fertilizer (F) (kg ha$^{-1}$ yr$^{-1}$) | 10 | 8 | 45 | | 10 | 8 | 45 | |

---

## Referee Report (RR2)

**Review of** *Current, steady-state and historical weathering rates of base cations at two forest sites in northern and southern Sweden: A comparison of three methods* **(v4, 2nd revision?) by Casetou-Gustafson et al. BG-2019-47R1 by Michael Velbel**

*The science*

1. No new comment – response to previous comment is satisfactory.

2. No new comment – responses to previous comment are satisfactory.

3. No new comment – response to previous comment is satisfactory.

4. No new comment – response to previous comment is satisfactory.

5. No new comment – response to previous comment left a bit to be desired.

6. You're welcome.

7. No new comment.

8. Previous comment: Weakness: The treatment of error is confusing. Eight plots were sampled, four in each of two study areas. At each study are, two control plots and two fertilized plots were sampled. Both fertilized plots at Flakaliden were "eliminated from further consideration in calculations of historical weathering rates using the depletion method" (lines 282 -283). Site mean or average values and their standard errors or combined standard uncertainty were calculated for each of the three approaches. "For the weathering rates based on the depletion method and the PROFILE model, error bars represent the standard error calculated based on four soil profiles at each study site, except for Flakaliden, where the depletion method was only applied in two soil profiles." (Lines 1045-1048, caption for Figure 4). However, error bars are shown for all cations for the depletion approach applied to Flakaliden; the reader is left to infer that these error bars show the range (not the SEs) of the rates estimated for the (only) two Flakaliden profiles. New comment: This reviewer still does not understand what the SE is for only two profiles.

9. No new comment – response to previous comment is satisfactory.

10. OK.

11. The reviewer is pleased.

12. The reviewer is pleased.

13. No new comment – response to previous comment is satisfactory.

14. You are welcome.

15. No new comment – response to previous comment is satisfactory.

16. No new comment – response to previous comment is satisfactory.

17. We are of like mind.

*The manuscript*

18. No new comment – response to previous comment is satisfactory.

Regarding the organization of the scientific content in the manuscript: Specific matters relating to this manuscript are highlighted in the following numbered items and the subsequent line-index list.

19. No new comment – response to previous comment is satisfactory.

20. No new comment – response to previous comment is satisfactory.

21. No new comment – response to previous comment is satisfactory.

22. No new comment – response to previous comment is satisfactory.

23. The courtesy to readers is much appreciated.

24. No new comment – response to previous comment is satisfactory.

25. No new comment – response to previous comment is satisfactory.

26. No new comment – response to previous comment is satisfactory.

27. No new comment – response to previous comment is satisfactory.

28. From here to #37, no new comment – responses to previous comments are satisfactory.

General comments:

There are many places in the text where, alone or in concert, insertion or redeployment of commas and semi-colons may make long sentences easier to read. Cannot ascertain how completely this was addressed.

Specific comments:

New comment: No caption for Figure 4 is provided in this version.

All reviewer suggestions were satisfactorily incorporated.

---

## Author Response (AR2)

**Authors response to reviewers**

We have only achieved the detailed review report by Prof Velbel, why we have based our revision on his comments and suggestions for improving the manuscript, beside the editors note. In addition to that, we have made additional corrections of flaws and poor design in the text, figures and tables when necessary. We thank the reviewer (Velbel) for having scrutinized the manuscript and providing a comprehensive, well structured and useful report that has helped us to improve the paper. The reviewer emphasizes three major points, which were also stressed by the Editor: (1) to strengthen the connection between problem definition, aims and conclusions, aims should be defined in a clearer way and novelty in approach and results clearly stated, (2) it should be written in a globally relevant perspective to attract the international scientific community, (3) it lacks important references (is under-referenced). Besides remarks on terminology, language and some flaws, as well as recommendations (include a map etc.), we notice that these critical remarks mostly concerns the Abstract, Introduction, Discussion, and Conclusions, but not the Material and methods, and Results sections.

This is in brief our response to the major critique: In the Introduction, we initially stress that the demands for more accurate estimate of base cation weathering rates emerged from soil and water acidification, and the useful concept of critical loads of acidity. This is a more logical entry and explains for example the development of the PROFILE model and the wide application of this model in e.g. Europe and North America. The rationale for the study is now expressed more clearly, leading to the new formulation of the aims, which is "to analyse the causes of discrepancies in estimations of weathering rates, with focus on conceptual versus random sources of discrepancies, between the depletion method, the PROFILE model and the balanced base cation budget approach." We think that an analysis of conceptual and random sources of discrepancies between methods is useful and novel in a general sense that can attract wider groups of the scientific community, although our approach may not be entirely new. In the Discussion, the subsections follow the three criteria and is updated to follow the aims. The introductory part of the Discussion is now deleted because the new Introduction made it redundant. The Conclusion is changed to make a stronger connection to the aims. A consequence of the revision is also that a number of references have been added, and a few have been deleted.

The changes listed below are numbered as by the reviewer (bg-2019-47-referee-report-3.pdf), and refer to line numbers in the revised manuscript.

1. This is a summary of the reviewers comments that we response to in detail below.

2. We deleted the sentence about MAGIC, because we deemed it not necessary in this background description. However, the MAGIC model has been used as an approach to estimate weathering rates based on measured and modelled ion fluxes. Weathering rate of different base cations is a parameter in MAGIC that can be set to achieve a balance between sources and sinks, and in this meaning it contains a budget approach. Some new references include important pioneering works on the budget method.

The sensitivty of PROFILE with respect to dissolution kinetics is considered and acknowledged in the Discussion and Conclusion. We have added a sentence about this on line 179-182 in the Introduction to make the connection between Introduction and later sections stronger.

4. OK, we can see the problem, but as the reviewer also acknowledge, we address this issue in detail later in the manuscript. The problems associated with the application of the depletion method are indeed important, and we stress this point further in the revised manuscript by introducing the concepts of conceptual and random sources to causes of discrepancies between method outputs. We think this is one nobelty of the paper.

5. Yes, BD was measured for a majority of soil samples.

6. Thanks, we apprepciate this note.

7. OK, we admit that this might be a problem, but we have also given fair rationales to why this was made, on Lines 489-494 in the revised manuscrcipt (no changes made).

8. OK, the Figure 4 caption can be misleading, so we have reformulated it. It now states that all error bars are SE, or the equivalent. Line 1097, Figure caption.

9. We agree. Previous work that shows closest resemblance for Na weathering, between depletion and budget calculations, are now referred to in the Discussion on line 706-711.

10. We think this comment is much included the abovementioned note (#9).

11. We appreciate this comment

12. We appreciate this comment.

13. We agree that Zr mobilityy and corrosion can be an issue for the depletion method, but we decided to not stress this question further in the Discussion due to lack of relevant data.

14. Thanks.

15. Thanks. See our comment to #9. We have also added some additional (new) references (Rosenstock 2019a, b, Callesen et al 2016) in support of our conclusions about the budget method: last paragraph of Discussion.

16. This comment of the contribution of the study to the scientific community is a very general statement. Our ambition with the revised manuscript is to overcome this critique, described elsewhere in our response. See the first paragraph of this report for a general answer.

17. This is compliment we take to our hearts.

18. This comment is a summary of recommended changes. Our changes to meet these requirements have been explained elsewhere in this response.

19. We agree fully! As described in the introductory part (2nd# paragraph) of this response letter, we have made major changes particularly in the Introduction to give a more clear background, explicit rationale of study (Line 125-141), and aims (L204-209), that we hope will be of significant for a wider group of readers. In particular, we introduce the concept of conceptual versus random causes to discrepancies between different methods to estimate weathering rates, and link these concepts to our ambition to make a harmonised comparison of methods. We have also made a number of changes in the Discussion to make a clearer connection to questions raised in the Introduction. Furthermore, we believe that one of the major strengths of our study is the focus on weathering and nutrient uptake by forest trees. Abstract and Conclusions have been changed in accordance with these changes.

20. The comment touches upon similar aspects (on regional vs global interest) as previous points made by the reviewer: Taken literally, a regional (local) study may be of local interest only, but it may also mean that it is of no general interest, in the sense of importance to scientific abstractions and theory, which by definition are rarely 'local'. However, most studies in this field of science are literally local as they are based on data from a limited number of sites, but can nevertheless be of general, theoretical value. The present study of course remains as a case study based on two forest sites about 1000 km apart, but we hope that the revised version will add more general value to the scientific community. This was achieved by a small shift in focus and stronger stress on analysing the causes of discrepancies between method, and by separating the conceptual from the random origins of discrepancies. The ideas behind the concepts have been latent, but are now explicit.

21. We hope the present revision of the Introduction, Discussion and Conclusions now meet those expectation. (Similar aspects as for #16, 19, 22)

22. This is not a review paper, but results from previous studies in Sweden using the same methods are referred to, as a way to see if our results are consistent with previuos studies. See our answer on #20 to the critique that this study is of local significance only. Also, notice how the aims are formulated in the revised manuscript.

23. We have included a map over Scandinavia, showing the locations of the sites.

24. Thanks, we realized that figures and tables in particularly the supplement were not in correct order, and has corrected that.

25. OK, we have corrected this flaw.

26. The reviewer points out areas of under-referencing, and we have accordingly added a few more references on the specific issue.

27. This is matter for editorial decision, which we will follow. Biogeosciences require that Supplements should be uploaded as pdf. In addition, BG also demands that source data should be available at DOI data repository sites. By the way, we have revised the Supplement with respect to order, legend and figure captions, and table design.

28. OK, corrected throughout the paper.

29. OK, corrected, line 673.

30. OK, corrected.

31. OK, corrected (L452)

32. OK, corrected. We write: 'due to their resistance to weathering'.

33. OK, corrected, we used the term 'constant' (Line 319) and 'uniform at L629, 631.

34. OK, soil layer changed to soil horizon throughout the manuscript, when relevant.

35. OK, corrected (Line 235, 430)

36. OK, corrected on numerous places.

37. OK, we have checked the references to harmonise with the requirements by the journal.

**Below: Velbels comments are in italics.**

*Specific comments:*

*Line 101 – replace "if" with "whether"*
- Whole sentence is changed
*Line 170 – replace "on" with "to"*
- Whole sentence is replaced

*Line 353 – replace "ar" with "is"*
- OK, corrected (Line. 405...)

*Line 416 – Delete close paren.*
- Done.

*Line 481 – Replace "as opposed" with "in contrast".*
- OK, corrected (Line 537)

*Line 580 – Replace "were possible to reconcile" with "could be reconciled"*
- OK, corrected (Line 663)

*Line 648 – delete the duplicate period.*
- OK, corrected (Line 700)

*Line 832 – "Sedimentologists" is plural.*
- OK, corrected (Line 884)

*Line 864 – Journal title should be in title case (all major words capitalized).*
- OK, corrected (Line 932)

*Figure axis labels should be in the format "Label (units)". The experienced reader presumes that elemental "concentration (%)" in Figure 2 means weight %, but, because it could be atomic or molar %, ("wt. %") would eliminate the possibility of misunderstanding by non-specialists and novice readers. The labeling of axes for all other Figures is excellent.*
- OK, corrected, new Figure # is Fig.3.

*Tables S1 and S2 are not useful as formatted. Graphical representation of the sensitivity analysis is required if it is intended to be understood by readers.*
- OK, we have made new graphs based on the tables, Figures S4-S5

*Tables S3 and S1b contain similar data for the two field areas; the numbering of these tables does not make sense.*
- Agree, the Supplement is revised.

*Table S4 – Reporting model-input soil bulk densities and exposed mineral surface areas to 15 significant figures is not justified by anything explicitly stated in the text.*
- Agree, that Supplement table has been trimmed.

*These comments, above and below, are intended to help improve the effective presentation of the work done and the scientific impact of the revised manuscript.*
Thank you!

[revised manuscript text omitted]

* * *
**Margin comments:**

Johan Stendahl 2019-10-19 16:46
**Kommentar [2]:** Perhaps this makes it more clear

Magnus Simonsson 2019-10-19 16:46
**Kommentar [3]:** I'm not sure what this means.

Bengt Olsson 2019-10-19 16:46
**Borttagen:** Our comparison of three approaches to estimate weathering rates showed significant discrepancies in spite of the fact that the study was well harmonised at the spatial scale and originated from analyses of the same soil material.

Bengt Olsson 2019-10-16 15:36
**Borttagen:** . . ... [5]

Bengt Olsson 2019-10-3 10:24
**Borttagen: 4.1 Comparison of conceptually different methods** . ... [6]

Magnus Simonsson 2019-10-15 16:54
**Borttagen:** Compared with previous studies, m

Bengt Olsson 2019-10-3 10:25
**Borttagen:** Modelled

Magnus Simonsson 2019-10-15 16:59
**Borttagen:** . ... [7]

Bengt Olsson 2019-10-3 10:37
**Borttagen:** of 4

Bengt Olsson 2019-10-16 14:04
**Borttagen:**

Bengt Olsson 2019-10-17 13:31
**Borttagen:** at

Bengt Olsson 2019-10-17 13:31
**Borttagen:** at

Bengt Olsson 2019-10-3 10:26

[revised manuscript text omitted]

Bengt Olsson 2019-10-3 10:31
**Borttagen: 3**

Bengt Olsson 2019-10-11 10:29
**Borttagen: S**

Bengt Olsson 2019-10-16 14:04
**Borttagen:**

Bengt Olsson 2019-10-16 14:04
**Borttagen:**

Magnus Simonsson 2019-10-15 18:03
**Borttagen: illustrated**

Magnus Simonsson 2019-10-15 18:04
**Borttagen: . This is**

Magnus Simonsson 2019-10-15 18:04
**Borttagen: by**

Magnus Simonsson 2019-10-15 18:04
**Borttagen: the study by**

Bengt Olsson 2019-10-19 16:55
**Borttagen: it**

Magnus Simonsson 2019-10-15 18:05
**Borttagen: conditions in**

Magnus Simonsson 2019-10-15 18:05
**Borttagen: of**

Bengt Olsson 2019-10-18 13:37
**Borttagen: indicated**

Magnus Simonsson 2019-10-15 18:07
**Borttagen: s**

Magnus Simonsson 2019-10-15 18:07
**Borttagen: of**

Bengt Olsson 2019-10-17 13:35
**Borttagen:** The Na fluxes differed f ... [10]

Bengt Olsson 2019-10-17 22:06
**Formaterat:** Inte Färgöverstrykning

Bengt Olsson 2019-10-17 21:59
**Borttagen: 5**

Magnus Simonsson 2019-10-15 18:25
**Kommentar [4]:**

Bengt Olsson 2019-10-17 13:35
**Borttagen: Since**

[revised manuscript text omitted]

Unknown
**Formaterat:** Typsnitt:Times, Fet

[Figure]

Figure 2.

Unknown
**Formaterat:** Typsnitt:Times, 10 pt

[Figure]

Figure 3

[Figure]

Figure 4.

[Figure]

Figure 5.

[Figure]

Figure 6.

[Figure]

Figure 7.